# PIN1-SUMO2/3 motif suppresses excessive RNF168 chromatin accumulation and ubiquitin signaling to promote IR resistance

Anoop S. Chauhan [1,2], Matthew J. W. Mackintosh[1,2,3], Joseph Cassar[2,3], Alexander J. Lanz[1,2], Mohammed Jamshad[1,2], Hannah L. Mackay[1,2], Alexander J. Garvin [1,2,4], Alexandra K. Walker[1,2], Satpal S. Jhujh [1,2], Teresa Carlomagno [2,3], Aneika C. Leney [2,3], Grant S. Stewart [1,2] ✉ & Joanna R. Morris [1,2] ✉

RNF168 is an E3 ubiquitin ligase critical to the mammalian DNA double-strand break repair response. The protein is recruited to and amplifies ubiquitin signals at damaged chromatin and, if not properly regulated, can drive an uncontrolled ubiquitin cascade potentially harmful to repair outcomes. Several indirect mechanisms restrict RNF168 positive feedback, and a long-standing question has been whether these alone suppress excessive RNF168 signaling or whether mechanisms to remove RNF168 from damaged chromatin exist. Here, we reveal a cascade of post-translational modifications which act at three adjacent amino acids, threonine-208, proline-209 and lysine-210, to process RNF168 actively. Phosphorylation at threonine-208 by CDK1/2 induces interaction with the peptidyl-prolyl isomerase PIN1. PIN1 promotes RNF168 SUMOylation at lysine-210, resulting in p97/VCP mediated removal. These actions promote RNF168 clearance and limit RNF168 chromatin build-up. Thus, single amino acid substitutions of the regulatory motif (SUMO-PIN1-assisted Chromatin Regulator, SPaCR) that restrict PIN1 interaction or SUMOylation are sufficient to drive supraphysiological accumulation of RNF168, increased ubiquitin signaling, excessive 53BP1 recruitment and radiosensitivity. Our findings define a mechanism of direct RNF168 regulation that is part of the normal damage response, promoting RNF168 dissociation from chromatin and limiting deleterious ubiquitin signaling.

The signaling cascade activated at chromatin by DNA double-strand breaks (DSBs) in mammalian cells depends on several post-translational modifications orchestrated by phosphorylation, ubiquitination and SUMOylation. Ubiquitination is driven by a series of E3 ubiquitin (Ub) ligases, initiated by the chromatin modification activities of RNF8 and RNF168[1–5]. RNF8 catalyzes K63-linked poly-ubiquitination of Histone H1[6] and chromatin-bound L3MBTL2[7], which triggers the recruitment of RNF168 via its Ub-binding motifs[1,5,8,9]. RNF168, in

[1]Department of Cancer and Genomic Sciences, School of Medical Sciences, College of Medicine and Health, University of Birmingham, B15 2TT Birmingham, United Kingdom. [2]Birmingham Centre for Genome Biology and Department of Cancer and Genomic Sciences, Medicine and Health, School of University of Birmingham, Birmingham, United Kingdom. [3]School of Biosciences, College of Life and Environmental Sciences, University of Birmingham, B15 2TT Birmingham, United Kingdom. [4]Present address: SUMO Biology lab, School of Molecular and Cellular Biology, Faculty of Biological Sciences, University of Leeds, LS2 9JT Leeds, United Kingdom. ✉e-mail: g.s.stewart@bham.ac.uk; j.morris.3@bham.ac.uk

turn, catalyzes K63-poly-ubiquitination and ubiquitination of Histone H2A and H2A variants[8,10–12]. RNF168-modified H2A is bound by 53BP1[13] and BARD1-BRCA1[14], which supports non-homologous end-joining (NHEJ) or homologous recombination (HR) repair of DNA DSBs, respectively (reviewed in ref. [15]). Biallelic mutations in *RNF168* cause RIDDLE syndrome (Radiosensitivity, ImmunoDeficiency, Dysmorphic features, and LEarning difficulties), highlighting the importance of the DNA damage-induced Ub response for normal development and immune system maturation[16].

Structurally, in addition to its RING domain responsible for its E3 ligase activity, RNF168 contains two Ub-binding domains; UDM1 (ubiquitin-dependent DSB recruitment module 1) and UDM2[8,11,12]. UDM1 recognizes Ub marks catalyzed by RNF8, whereas UDM2 binds ubiquitinated-H2A/H2AX and contributes to the amplification of RNF168 recruitment to damaged chromatin[6,8,11]. Mechanisms that limit RNF168 or its Ub signaling are thus vital for restraining excessive RNF168 accumulations and the subsequent dysregulation of downstream pathways. RNF168 protein restriction is a critical limiting feature of Ub-dependent DNA damage signaling and loss of proteins that curb RNF168 protein expression levels (mTORC1–S6K and TRIP12/UBR5), result in its excessive accumulation at the sites of DNA damage[17,18]. Similarly, increased RNF168 expression through chromosome amplification in some cancers results in an exaggerated Ub-signaling, deregulated sequestration of downstream proteins and altered DSB repair[19]. RNF168-Ub-signaling is also limited by interactions that suppress RNF168 E3 ligase activity[20], the activity of its partner E2 ubiquitin-conjugating enzyme[21], or through the processing of ubiquitylated chromatin by deubiquitinating enzymes[22–29]. Given the potency and nodal position of RNF168 in the DNA damage response, a longstanding question has been whether such indirect mechanisms are sufficient to explain the suppression of excessive RNF168 spreading and signaling or whether an active, as yet unknown, pathway exists to remove it from chromatin.

Here, we identify a previously unknown regulatory motif in RNF168, the SUMO-PIN1-assisted Chromatin Regulator, SPaCR, and delineate the post-translation cascade that acts upon it. We demonstrate that the threonine residue within the SPaCR motif is phosphorylated by CDK1/2, that phosphorylation then regulates interaction with the phosphorylation-dependent peptidyl-prolyl isomerase, PIN1, which in turn promotes RNF168 SUMOylation and ubiquitination, in turn driving p97/VCP mediated clearance. Consequently, mutation of the SPaCR motif allows unrestricted accumulation of RNF168 on damaged chromatin, which sequesters Ub-sensitive DNA repair proteins such as 53BP1, and subsequently compromises the repair of ionizing radiation-induced DNA breaks. These results uncover a direct mechanism cells use to regulate RNF168 in the damage response.

## Results

### PIN1 regulates RNF168 chromatin retention

Since RNF168 can bind to and amplify Ub conjugates generated by its own activity, we hypothesized that mechanisms beyond modulating protein stability or chromatin ubiquitination might be required to constrain its spread. We considered that direct post-translational modification (PTM) of RNF168 could represent one possible mechanism to regulate its activity and mined the publicly available "eukaryotic linear motif (ELM) resource"[30] for potential RNF168 regulatory motifs. Intriguingly, we found numerous potential PIN1-binding sites within and near the Ub-binding domains (UDMs) of RNF168 (Supplementary Fig. 1A).

To examine if PIN1 impacts RNF168, we examined the focal recruitment of endogenous RNF168 in untreated and irradiation (IR)-treated cells following PIN1 depletion. Notably, cells lacking PIN1 displayed an increased intensity of RNF168 at sites decorated with phosphorylated H2AX, γH2AX, when compared to control-treated cells in both untreated and IR-treated conditions (Fig. 1A, B and

Supplementary Fig. 1B–C). When we fractionated cell lysates, we observed increased RNF168 in the chromatin fraction of PIN1-depleted cells, which was evident in untreated cells but more prominent after IR (Fig. 1C). Whilst RNF168 protein levels appeared marginally higher in PIN1-depleted cells this is insufficient to explain the observed excess accumulation on chromatin (Fig. 1C Supplemental Fig. 1 D). Similarly, we saw a dramatic increase in the chromatin association of exogenous myc-tagged RNF168 after PIN1 depletion (Fig. 1D–H and Supplementary Fig. 1D). Interestingly, the accumulation of myc-RNF168 in unirradiated, PIN1-depleted cells extended beyond the boundaries marked by γH2AX (Fig. 1D), which was reminiscent of previous findings demonstrating the supra-physiological RNF168 recruitment to chromatin when protein quality control is suppressed[17]. However, in contrast, the increased accumulation of myc-RNF168 on chromatin in PIN1-depleted cells was not associated with alteration in the total cellular pool of RNF168 (Fig. 1H). To investigate whether PIN1 activity influences RNF168, we tested a series of PIN1 inhibitors, two targeting its enzymatic activity (Juglone[31] and PiB[32]) versus one targeting its stability (ATRA[33]). Cells treated with all three inhibitors exhibited increased RNF168 accumulation intensities in untreated and IR-treated cells, indicating that PIN1 activity is important for regulating the levels of chromatin-bound RNF168 (Fig. 1I–J and Supplementary Fig. 1E).

### Threonine-208 phosphorylation promotes PIN1 interaction

PIN1 acts to isomerize prolyl bonds between proline and the preceding amino acid. It is unique among PPIases as it also has an N-terminal "WW" domain that targets the enzyme to substrates containing pS/pT-P motifs[34–37]. To address whether the WW domain binds RNF168, we generated and purified wild-type (WT) GST-PIN1-WW domain and a mutant form, GST-WW-W34A, unable to bind phosphorylated serine or threonine residues[38]. We found the WT-WW but not the W34A mutant efficiently co-precipitated RNF168 from U2OS and HeLa cells (Fig. 2A, B). Moreover, we found that the binding of RNF168 to the PIN1-WW domain was enhanced when cells were irradiated (Fig. 2C).

Three potential PIN1 binding motifs are located in amino acids (aa 190–235), adjacent to UDM1 of RNF168. No function has been attributed to this region, so to investigate whether it bears elements important to RNF168 function, we first tested whether it is required for cellular resistance to IR. As expected, the depletion of RNF168 sensitized cells to IR, which could be rescued by the reexpression of WT RNF168. Importantly, the expression of an RNF168 mutant lacking aa 190–235 failed to rescue resistance to IR (Fig. 2D–F), suggesting this region may bear sequences relevant to RNF168 function. We next addressed whether any of the three potential PIN1 binding motifs within this region (S197/P198, T208/P209 or T230/P231) were required for the interaction of RNF168 with the PIN1 WW-domain. We mutated each serine/threonine to alanine and found that the T208A mutation, but not S197A or T230A, compromised the interaction of RNF168 with the GST-WW-PIN1 domain (Fig. 2G & Supplementary Fig. 2A).

To test whether the phosphorylation of T208 can mediate the interaction with PIN1 directly, we incubated phosphorylated and unphosphorylated RNF168 peptides corresponding to aa S204-K210 with full-length PIN1 and carried out native mass spectrometry. Direct peptide binding will increase the molecular mass and charge-state distribution of PIN1, which the mass spectrometry can detect. Using this technique, we observed that PIN1 preferentially bound the phospho-RNF168 peptide (Fig. 2H), suggesting that phosphorylation of RNF168 on threonine 208 facilitates its interaction with PIN1.

### Threonine-208 of RNF168 is a target of CDK1/2 kinases

Phosphorylation of RNF168 threonine 208 has been identified in bulk phospho-proteome analysis[39]. As a tool to identify the kinase(s) responsible for the modification, we generated an antibody against a phosphorylated RNF168 peptide (SDPV(pThr)PKSEKKSKNC). This antibody recognized WT-RNF168 in cell lysates, but not T208A-RNF168

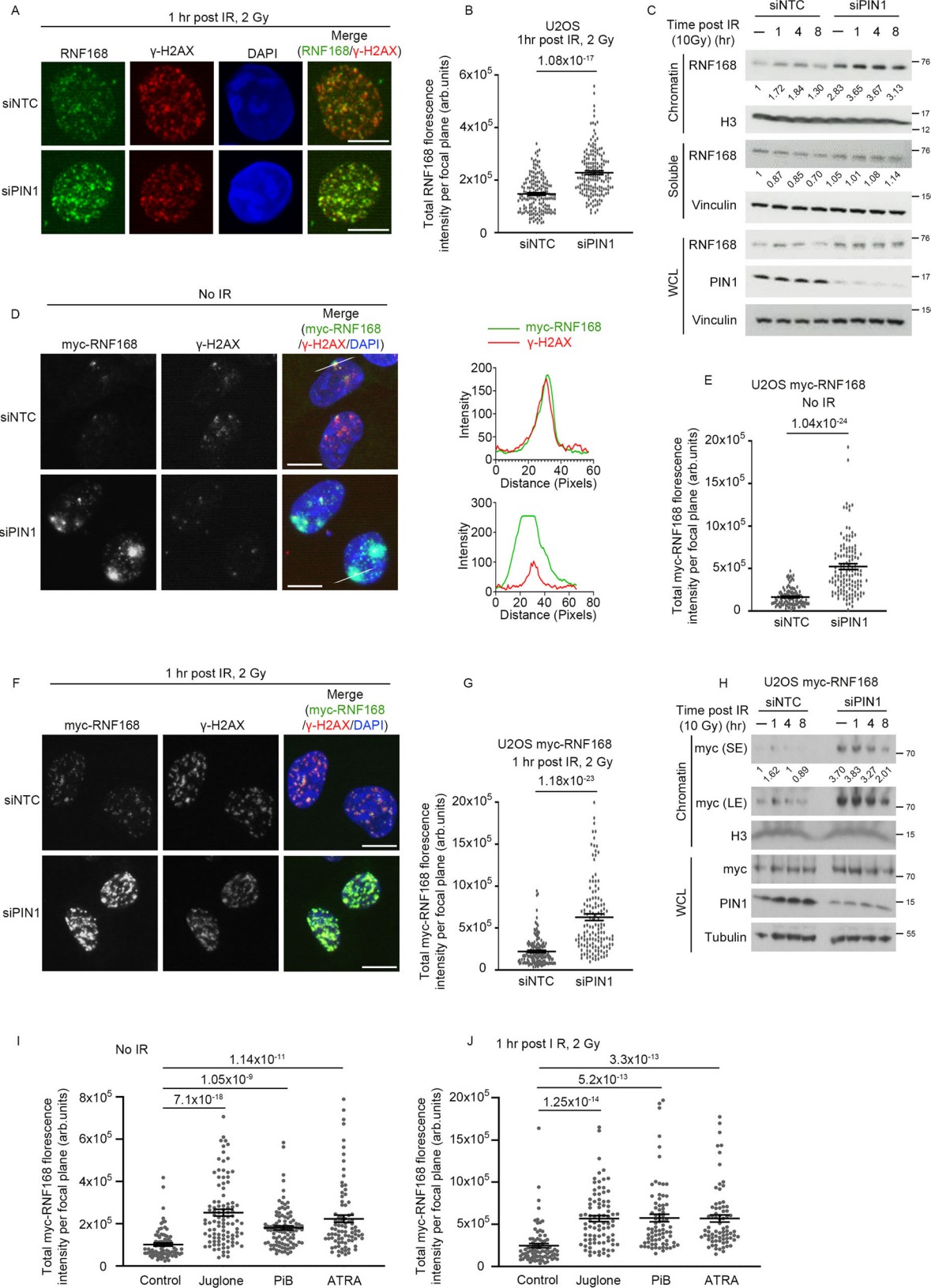

(Fig. 2I). To identify which kinase(s) are responsible for RNF168 phosphorylation, we tested several inhibitors, including those targeting proline-directed kinases. Notably, treatment with the CDK inhibitors roscovitine and R03306 significantly reduced T208-RNF168 phosphorylation (Fig. 2J and Supplemental Fig. 2B). Consistent with this, the depletion of CDK1/2, but not JNK1/2 or GSK3, also reduced

T208 phosphorylation (Fig. 2K). Moreover, we observed that GFP-RNF168 could co-precipitate both CDK1 and CDK2 from cell extracts (Supplemental Fig. 2C). We next tested the prevalence of pT208 in cells released from an S-phase double thymidine-block without DNA damage, noting that pT208 levels peaked 4–8 h after release and before the appearance of pS10-histone-3 (Supplemental Fig. 2D), a

**Fig. 1 | PIN1 regulates RNF168 chromatin retention. A** Representative images of RNF168 foci in control (siNTC) or PIN1-depleted cells (siPIN1) after treatment with ionizing radiation (IR), 2 Gy. Scale bars 10 µm. **B** Quantification of RNF168 foci intensity from A. Data is mean ± s.e.m, $n = 170$ cells. Source data are provided as a Source Data file. **C** Western blot of chromatin and soluble fractions or whole cell lysate (WCL) for RNF168, histone H3, vinculin and PIN1. Control and PIN1 depleted U2OS cells were treated with 10 Gy of IR and collected after indicated time points to prepare chromatin, soluble fraction or WCL (performed once). Source data are provided as a Source Data file. **D** Representative images of myc-RNF168 foci in control (siNTC) or PIN1-depleted cells (siPIN1). Scale bars 10 µm. The right panel shows the fluorescence intensity profiles along the line for myc and γH2AX. **E** Quantification of myc-RNF168 foci intensity from (**D**) Data is mean ± s.e.m, $n = 113$ cells for siNTC and 131 cells for siPIN1. Source data are provided as a Source Data file. **F** Representative images of myc-RNF168 foci in control (siNTC) or PIN1-depleted cells (siPIN1) after 2 Gy IR. Scale bars 10 µm. **G** Quantification of myc-RNF168 foci intensity from (**F**). Data is mean ± s.e.m, $n = 142$ cells for siNTC and 136 cells for siPIN1. Source data are provided as a Source Data file. **H** Western blot of chromatin fraction and whole cell lysate (WCL) for myc, histone H3, PIN1 and tubulin in control (siNTC) or PIN1-depleted cells (siPIN1) after treatment with 10 Gy IR. SE: short exposure, LE: Long exposure (Representative of 2 repeats). Source data are provided as a Source Data file. **I** Quantification of myc-RNF168 intensity after treatment with various PIN1 inhibitors: Juglone (10 µM, 4 hrs), PiB (25 µM, 24 hrs), ATRA (25 µM, 24 hrs). Data is mean ± s.e.m, $n = 82$ for control, 100 for Juglone and PiB, and 86 for ATRA. Source data are provided as a Source Data file. **J** Quantification of myc-RNF168 foci intensity after treatment with PIN1 inhibitors and IR. Cells expressing myc-RNF168 were treated with Juglone (10 µM, 4 hrs), PiB (25 µM, 24 hrs), ATRA (25 µM, 24 hrs) before irradiation (2 Gy IR). Data is mean ± s.e.m, $n = 82$ for control, 84 for Juglone, 77 for PiB, and 74 for ATRA. Source data are provided as a Source Data file.

timing consistent with late S-G2 and with the reported peak activity of CDK kinases[40]. These findings suggest phosphorylation of RNF168 at threonine 208 is catalyzed by CDK1/2 kinases.

## Manipulation of RNF168 threonine-208 and proline-209 alters RNF168 chromatin accumulation

Since pT208 drives PIN1 interaction, we predicted that mutation of this residue would stimulate RNF168 chromatin accumulation in a manner similar to PIN1 inhibition or depletion. Indeed, when we examined the focal relocalization of the T208A-RNF168 mutant following treatment with IR, we found it formed brighter IR-induced RNF168 foci and showed more chromatin enrichment than WT-RNF168 (Supplementary Figs. 3A–C). In contrast, mutants inactivating the alternative putative PIN1 binding sites, S197A-RNF168 or T230A-RNF168 had no impact (Supplementary Fig. 3A, B and D). Importantly, RNF168 protein expression levels were unaffected by the T208 mutation, indicating that the increased chromatin binding is not a consequence of elevated RNF168 protein stability (Supplementary Fig. 3C). The mobilization of T208A-RNF168 to γH2AX-decorated chromatin was suppressed by RNF8 depletion (Supplementary Figs. 3E and F), demonstrating its recruitment retains dependency on the canonical Ub-dependent DNA damage signaling pathway.

In polypeptide chains, most peptide bonds between amino acids adopt a *trans* conformation[41]. Proline has a unique side chain, forming a five-membered ring, that allows the formation of both *cis* and *trans* prolyl bonds[42]. To test whether PIN1 can catalyze the isomerization of the region around T208-P209, we conducted a $^1H$-$^1H$ EXchange SpectroscopY (EXSY) using a short phosphorylated RNF168 peptide, SDPV(pT)PK in the presence and absence of PIN1. $^1H$-$^1H$ EXSY is a highly sensitive technique that can be used to detect the rapid exchange between *cis* and *trans* proline species[43]. When the exchange is slower than the detection timescale of the EXSY experiment, only diagonal peaks are observed for the *cis* and *trans* species. If the *cis-trans* isomerization rate is increased to become detectable by the EXSY experiment, characteristic off-diagonal cross-peaks, which correspond to the interconversions of *cis-trans* and *trans-cis*, appear. Using this method, we observed no cross-peaks for the pT208-RNF168 peptide alone at 300 ms mixing time (Fig. 3A), demonstrating a slow rate of *cis-trans* isomerization. However, following the addition of PIN1, characteristic cross-peaks appeared (Fig. 3A), indicative of increased *cis-trans* isomerization. The rate of interconversion ($k_{ex}$) calculated in the presence of PIN1 was $12.65 ± 0.18 \text{ s}^{-1}$ (Fig. 3A). Thus, PIN1 can catalyze the isomerization of the SDPV(pT)PK peptide.

To explore whether changes at proline-209 can impact the cellular behavior of RNF168, we mutated proline-209 to alanine, reasoning that this substitution increases the probability of a *trans*-isomer[44,45]. We compared the localization intensity of T208A-RNF168, P209A-RNF168 single mutants and the double mutant, T208A-P209A-RNF168, with that of WT-RNF168. The P209A-RNF168 mutant showed levels of localization and chromatin enrichment comparable to WT-RNF168. Strikingly, and in stark contrast to the T208A mutant, the double, T208A-P209A, mutant also exhibited levels of localization intensity and chromatin enrichment similar to WT, both in untreated and irradiated cells (Fig. 3B–E). These findings show the P209A mutation negates the harmful impact of the T208A mutation, and indicates that presenting a *trans*-favored conformation of RNF168 bypasses the need for T208 phosphorylation in suppressing excessive RNF168 chromatin accumulation.

## PIN1 activity is linked to RNF168 SUMOylation

RNF168 is reported to be SUMOylated[46,47]. Intriguingly, the PIN1 binding site at threonine-208 and proline-209 is immediately adjacent to a VTPKSE SUMO modification reported at lysine-210 [47–49]. Based on this, we hypothesized that pT208-dependent, PIN1-mediated isomerization of RNF168 might alter its SUMOylation status.

We first examined if lysine-210 is needed for SUMO2 modification, by expressing WT-RNF168 and K210R-RNF168 together with His-Flag-SUMO2, followed by nickel-enrichment of SUMO conjugates under denaturing conditions. Using this approach, we observed that WT-RNF168 was enriched and showed a band size consistent with mono-SUMOylation. In contrast, the K210R-RNF168 mutant was not enriched (Fig. 4A), indicating that lysine-210 is required for a large proportion of RNF168 SUMOylation. These data are consistent with a recent report that found lysine-210 is the main SUMO acceptor site in both cellular RNF168 and following in vitro SUMOylation[47].

To explore whether SUMOylation and PIN1 activity are connected, we precipitated His-Flag-SUMO2 from WT and PIN1-depleted cells and monitored the levels of RNF168-SUMO2 conjugates. The depletion of PIN1 decreased the detection of SUMO2-modified RNF168 (Fig. 4B). Similarly, mutation of threonine-208 to alanine also reduced RNF168 detected in SUMO conjugates (Fig. 4C). Notably, the presence of RNF168 in SUMO conjugates could be restored to WT levels when the P209A substitution was present in addition to T208A (Fig. 4C), indicating the P209A mutation negates the suppression of SUMOylation by the T208A mutation.

To assess whether SUMOylation regulates RNF168 chromatin accumulations, we depleted SUMO1 and SUMO2/3 (SUMO2 and SUMO3 isoforms share 97% sequence identity) and monitored RNF168 localization to sites of DNA damage. We observed that depletion of SUMO2/3, but not SUMO1, dramatically increased RNF168 chromatin enrichment and accumulation to γH2AX in undamaged and IR-treated cells (Fig. 4D–G and Supplementary Fig. 4A and B). Moreover, we also noted that mutation of the SUMO acceptor site K210R-RNF168, showed an increase in chromatin enrichment similar to that of the T208A-RNF168 mutant (Fig. 4H). These data are consistent with the idea that K210 SUMOylation suppresses excessive RNF168 chromatin association. Next, we wanted to address whether the inclusion of P209A, can suppress excessive accumulation of K201R-RNF168, as it

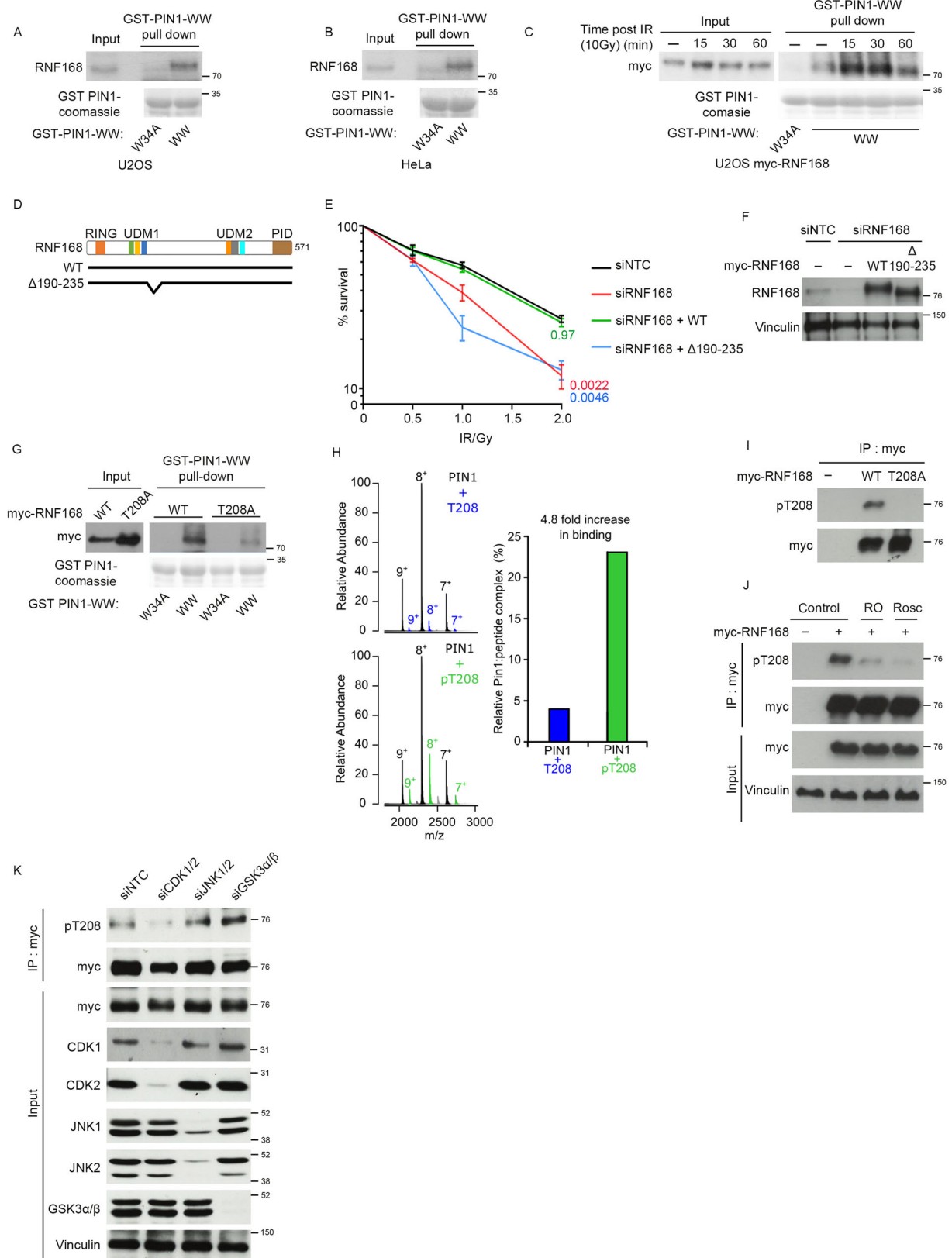

can for T208A-RNF168. When we assessed the chromatin association of the P209A-K210R-RNF168 double mutant, we found its levels of association were similar to that of the T208A- or K210R- RNF168 mutants, and greater than P209A-RNF168 or WT proteins (Supplementary Fig. 4C). These data demonstrate that K210R is dominant over P209A and suggest that the requirement for K210 in suppressing the

excessive accumulation of RNF168 on chromatin remains even when the *trans*-biased 209 codon is present. We have named this region a SPaCR motif for 'SUMO-PIN1-assisted Chromatin Regulator' to draw together the PIN1 dependency of SUMOylation and the requirement for each portion of the regulation in suppressing excessive RNF168 chromatin accumulation.

**Fig. 2 | Phosphorylated threonine-208 promotes PIN1 interaction. A** Pull down of endogenous RNF168 by GST-fused WW or W34A mutant domain of PIN1 from U2OS cells (Representative of 2 repeats). Source data are provided as a Source Data file. **B** Pull down of endogenous RNF168 by GST-fused WW or W34A mutant domain of PIN1 from HeLa cells (Representative of 2 repeats). Source data are provided as a Source Data file. **C** Pull down of myc-RNF168 by GST-fused WW or W34A mutant domain of PIN1 from U2OS cells treated with 10 Gy of IR. Cell lysates were made at indicated time points after IR (Representative of 2 repeats). Source data are provided as a Source Data file. **D** Schematic of RNF168 domains and deletion mutant. **E** Colony survival of U2OS cells depleted of RNF168 and complemented with WT, Δ190-235 RNF168 after treatment with indicated doses of IR. $n = 3$. Data is mean ± s.e.m. Source data are provided as a Source Data file. **F** Western blot to show depletion of RNF168 and complementation of RNF168 variants for (**E**) (performed once). Source data are provided as a Source Data file. **G** U2OS cells expressing myc-RNF168 wild type (WT) or T208A mutant were subjected to pull down by GST-fused-WW or W34A domain of PIN1 (Representative of 3 repeats). Source data are provided as a Source Data file. **H** Native mass spectrum of recombinant PIN1 incubated with peptides of the T208 motif from RNF168 at a 1:5 molar ratio (left). Colored peaks indicate PIN1:peptide complex formation between non-phosphorylated peptide (blue) and phosphorylated peptide (green), while black peaks correspond to unbound PIN1. Quantification of the PIN1:peptide peak intensities as a percentage of bound relative to total protein (right). Source data are provided as a Source Data file. **I** myc-WT-RNF168 or myc-RNF168-T208A were precipitated from HEK293 cells using myc-sepharose beads and probed with anti-myc or anti-phosphorylated-threonine-208 RNF168 (pT208) antibodies (Representative of 2 repeats). Source data are provided as a Source Data file. **J** HEK293 cells expressing myc-RNF168 were treated with RO-3306 (10 μM) "RO" or roscovitine (25 μM), "Rosc" for 4 hrs and probed with anti- pT208, myc and vinculin antibodies (performed once). Source data are provided as a Source Data file. **K** Assessment of pT208 and myc following immunoprecipitation of myc-RNF168 from control siRNA-treated cells (siNTC) and from cells treated with siRNAs to the kinases shown (Representative of 2 repeats). Source data are provided as a Source Data file.

## Threonine-208 promotes RNF168 clearance via and p97/VCP

Next, we addressed the relationship between SPaCR suppression of excessive RNF168 chromatin accumulation and the TRIP12/UBR5 suppression pathway. The E3 ubiquitin ligases, TRIP12, and UBR5 form part of the protein quality control network that acts to suppress RNF168 protein levels[17]. As expected, co-depletion of TRIP12/UBR5 increased the total cellular pool of WT-RNF168 and increased RNF168 present on chromatin. Notably, TRIP12/UBR5 suppression also increased T208A-RNF168 total protein levels and association on chromatin, suggesting TRIP12/UBR5 and SPaCR motif work independently to regulate the stability and chromatin association of RNF168 (Supplemental Fig. 5A). Interestingly, the levels of chromatin-associated RNF168 in TRIP12/UBR5 depleted cells did not differ between WT and T208A-RNF168, indicating that TRIP12/UBR5-dependent regulation of RNF168 dominates over its regulation by SPaCR motif.

The SUMOylation of chromatin-bound proteins has been associated with their interaction with SUMO-targeting E3 ubiquitin ligases and extraction from chromatin by the AAA+ ATPase p97/VCP[50–57]. To investigate whether PIN1-dependent SUMOylation of RNF168 affected its ubiquitination and subsequent extraction from chromatin by p97/VCP, we first assessed the impact of T208A and K210R mutation on RNF168 ubiquitination levels. Consistent with the SUMOylation of RNF168 being a pre-requisite for its ubiquitination, we observed that both the T208A and K210R, mutations reduced RNF168 ubiquitination. However, since K210 could be modified by either SUMO or ubiquitin[47,49,58], we cannot discount that the reduction of K210R-RNF168 ubiquitination may be independent of SUMOylation.

Therefore, to investigate whether RNF168 SUMOylation is required for its subsequent ubiquitination, we tested whether RNF168 interacts with the SUMO-targeting E3 ubiquitin ligase, RNF4. Consistent with RNF168 SUMOylation being required for its ubiquitination, we observed that WT-RNF168 interacted with RNF4 and that this interaction was compromised by preventing T208 phosphorylation (Fig. 5B). Next, we tested whether RNF4 influences the localization of RNF168 on chromatin. Notably, we observed that RNF4 depletion increased the association of WT-RNF168 but not the T208A mutant with chromatin, suggesting that loss of RNF168 T208 phosphorylation is epistatic with RNF4 knockdown (Fig. 5C). Interestingly, depletion of RNF4 had less of an impact on RNF168 chromatin association than the T208A mutation, indicating the presence of additional factors that contribute to the suppression of RNF168 turnover.

p97/VCP acts early in the DSB response regulating DNA repair protein recruitments and dissociations from chromatin[54,59–61]. Therefore, to address whether p97/VCP also functions to regulate the chromatin association of RNF168, we initially investigated whether RNF168 and p97/VCP could associate in cells. In support of a biochemical link between RNF168 and p97/VCP, we observed that GFP-RNF168 could precipitate p97/VCP from cell extract (Fig. 5D) and that treatment of irradiated cells with the p97/VCP inhibitor, CB-5083, increased focal intensity of RNF168 comparable to that observed with the T208A mutation (Fig. 5E, F). In contrast, p97/VCP inhibition had no impact on the intensity of IR-induced myc-T208A-RNF168 foci (Fig. 5E, F). Thus, p97/VCP inhibition and loss of RNF168 phosphorylation are also epistatic.

In many previous assessments of the role of p97/VCP in supporting DNA repair, p97/VCP inhibitors or siRNA have been added before IR-treatment, e.g.[59,61]. To address whether p97/VCP has a significant role later in the response, we reproduced the timeline applied to the RNF168 observations, adding p97/VCP inhibitor an hour after IR exposure. We found that this too was deleterious to cell survival (Supplemental Fig. 5C), suggesting a role for p97/VCP later in the response to IR.

## Threonine-208 restrains chromatin ubiquitination and 53BP1 recruitment

One possible consequence of increased RNF168 chromatin retention is increased chromatin modification. We assessed the ubiquitylation status of H2A/H2AX and found that cells complemented with the T208A-RNF168 mutant showed increased H2A ubiquitylation after IR as detected by substrate-specific H2AK15-Ub antibody (Fig. 6A). Similarly, we also saw an increase in the mono and di-Ub of γH2AX after IR in cells complemented with T208A-RNF168 mutant compared to cells expressing WT-RNF168 (Supplementary Figs. 6A–C). Ubiquitination of H2A/X at its N-terminus mediates 53BP1 binding[12,13,62,63]. When we examined the recruitment of 53BP1 to chromatin in T208A-RNF168 complemented cells, we observed enlarged and more intense 53BP1 foci at sites of γH2AX-decorated chromatin before and after irradiation (Fig. 6B–D). These data suggest that increased chromatin-associated RNF168 drives enhanced ubiquitin signaling and increased sequestration of 53BP1.

Since RNF168-dependent H2A/H2AX ubiquitination is also known to mediate interaction with BARD1 of the BRCA1:BARD1 heterodimer[14,64], we next examined BRCA1 recruitment to sites of DNA damage. As expected, the depletion of endogenous RNF168 compromised BRCA1 foci formation, which was rescued by siRNA-resistant WT-RNF168 (Fig. 6E, F). However, surprisingly, neither the T208A nor K210R RNF168 mutants (both of which ablate RNF168 SUMOylation) fully support BRCA1 recruitment to DSBs in CENPF-positive (G2) cells (Fig. 6E, F). These observations are consistent with previously published data demonstrating that PIN1 depletion compromises the ability of cells to form BRCA1 foci in response to IR[65].

PALB2 can stimulate RAD51 recruitment through its ability to bind to ubiquitin-bound RNF168[66], so we considered that excessive RNF168 chromatin binding might promote RAD51 accumulation at sites of

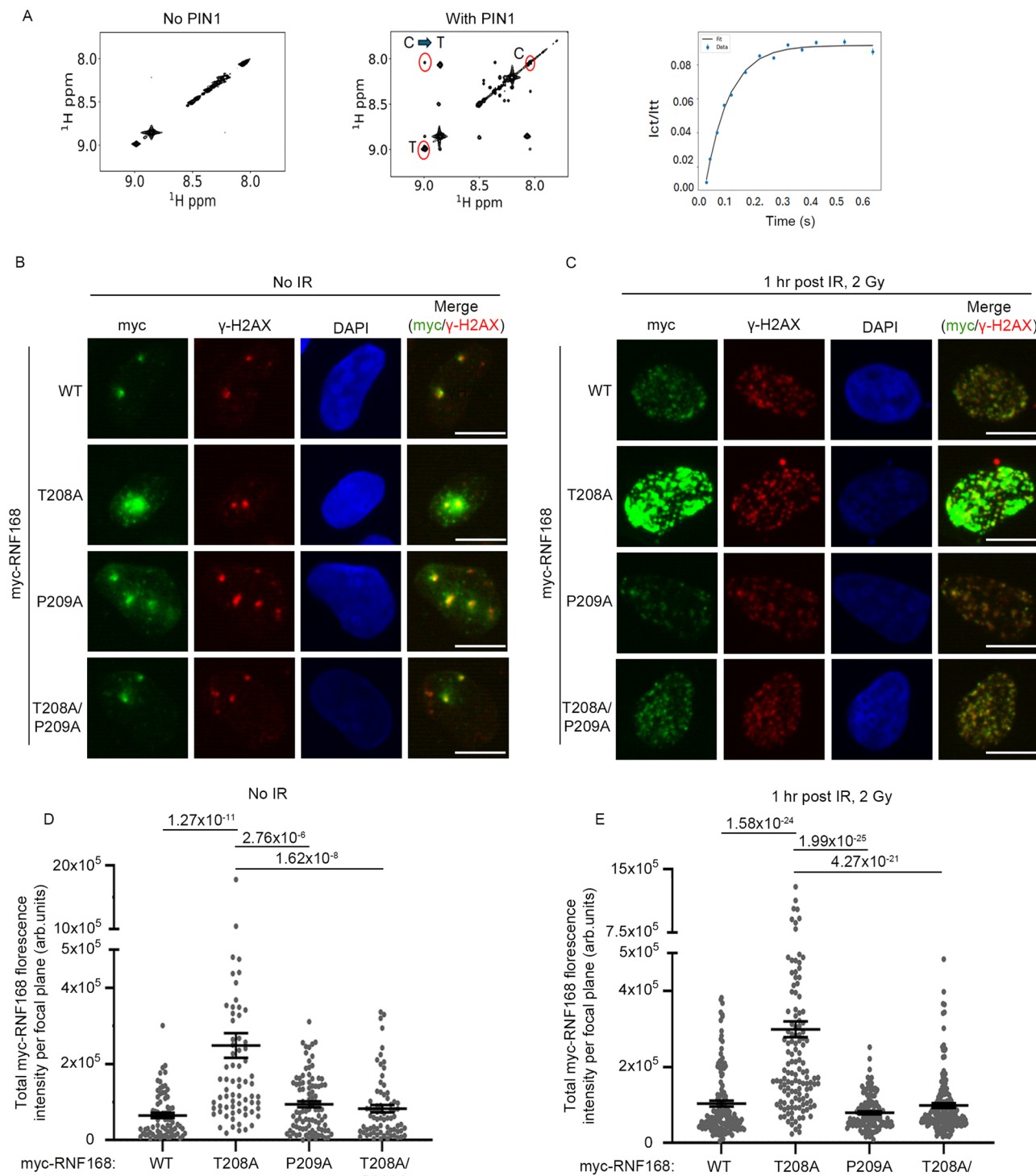

**Fig. 3 | P209A mutation rescues T208A-RNF168 hyper-accumulation. A** $^1$H−$^1$H EXchange SpectroscopY (EXSY) detected activity of SDPV(pT)PK RNF168 peptide with and without PIN1. Left: 2D $^1$H-$^1$H EXSY spectrum of 2 mM peptide in the absence of PIN1 (mixing time, 300 ms). Middle: 2D $^1$H-$^1$H EXSY spectrum of 2 mM peptide in the presence of 25 μM PIN1 (mixing time was 300 ms). T and C indicate the diagonal peaks when the proline is in either the trans or cis conformation, respectively. Right: $^1$H-$^1$H EXSY curves are plotted as the intensity ratio of the C to T peak over the T peak versus mixing times ranging from 12.5 ms to 600 ms and fitted as described in methods. The fitted value of kex is 12.65 ± 0.18 s-1. **B** Representative images of foci formation of RNF168 mutants. U2OS cells stably expressing siRNA resistant myc-WT-RNF168, T208A, P209A and T208A/P209A mutants. Cells were fixed and stained for myc and γH2AX. Scale bars 10 μm. **C** As in **B**, cells expressing myc-WT-RNF168, T208A, P209A, T208A/P209A mutants were treated with 2 Gy of IR and fixed after 1 hr. Cells were stained for myc and γH2AX. Scale bars 10 μm.
**D** Quantification of myc-RNF168 protein foci intensity from (**B**). Data is mean ± s.e.m, *n* cells= 73 for WT, 75 for T208A, 92 for P209A and 73 for T208A/P209A. Source data are provided as a Source Data file. **E** Quantification of myc-RNF168 protein foci intensity from C. Data is mean ± s.e.m, *n* cells = 152 for WT, 142 for T208A, 130 for P209A and 163 for T208A/P209A. Source data are provided as a Source Data file.

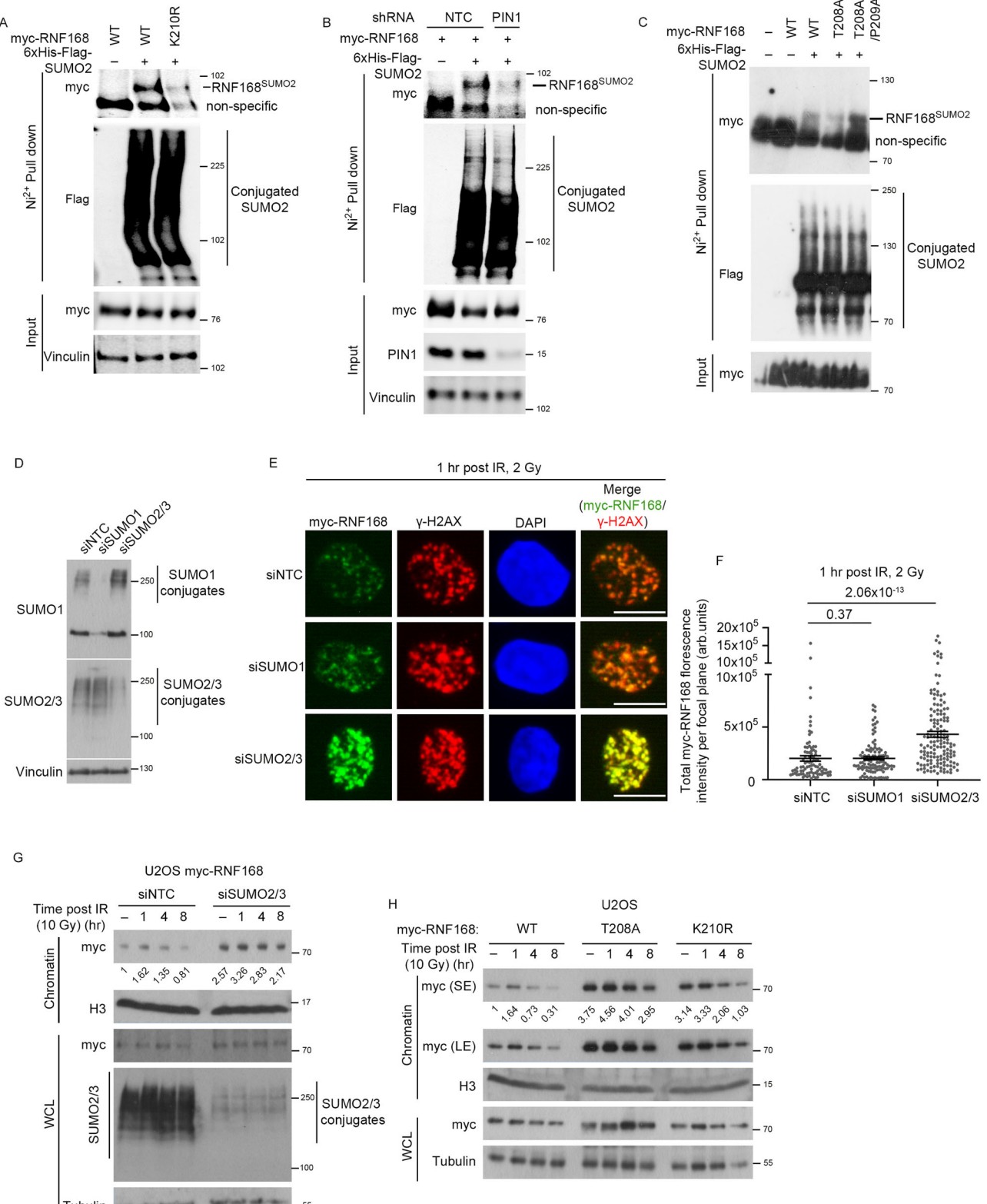

damaged DNA. However, neither T208A-RNF168 nor K210R-RNF168 supported RAD51 foci formation to the level observed with WT-RNF168 (Fig. 6G and Supplementary Fig. 6D). Thus, while chromatin ubiquitination is increased in T208A-RNF168-expressing cells, this selectively enhances the recruitment of 53BP1 to sites of DNA damage while suppressing the relocalization of the BRCA1:BARD1 heterodimer. To query the relationship

between PIN1 and BRCA1, we investigated their role in supporting survival after IR, finding that depletion of either suppressed cell survival and that co-depletion reduced survival no more than each individual depletion (Supplemental Figs. 6E, F). These data suggest that PIN1 and BRCA1 contribute to survival after IR through the same pathway, consistent with the observations of reduced BRCA1 recruitment.

**Fig. 4 | PIN1 promotes SUMO2/3ylation of RNF168 and SUMOylation regulates RNF168 chromatin accumulation. A** HEK293 cells complemented with myc-WT-RNF168 or K210R mutant were transfected with 6xHis-Flag-SUMO2. SUMO2 conjugated proteins were enriched by His-Mag Sepharose Ni beads (Ni$^{2+}$ pull down) under denaturing conditions and detected by western blotting (Representative of 2 repeats). Source data are provided as a Source Data file. **B** HEK293 cells containing shRNA control (NTC) or shPIN1 were transfected with myc-RNF168 alone or in combination with 6xHis-Flag-SUMO2. Cells were treated with 1 mM IPTG for 72 hrs for PIN1 depletion. SUMO2 conjugated proteins were enriched by His-Mag Sepharose Ni beads (Ni$^{2+}$ pull down) under denaturing conditions and detected by western blotting (Representative of 2 repeats). Source data are provided as a Source Data file. **C** HEK293 cells transfected with myc-RNF168-WT, T208A or T208A/P209A along with 6xHis-Flag-SUMO2. SUMO conjugated proteins were pulled down by His-Mag Sepharose Ni beads (Ni$^{2+}$ pull down) under denaturing conditions. SUMO-conjugated RNF168 variants were detected by western blotting (Representative of 2 repeats). Source data are provided as a Source Data file. **D** Western blot of SUMO1 and SUMO2/3 conjugates following siRNA treatments (performed once). Source data are provided as a Source Data file. **E** U2OS cells stably expressing myc-WT-RNF168 were treated with indicated control and SUMO siRNAs, cells were treated with 2 Gy of IR and fixed after 1 hr post IR and stained for myc and γH2AX. Scale bars 10 μm. **F** Quantification of myc-RNF168 intensity from (**E**). Data is mean ± s.e.m, $n = 90$ cells for siNTC, 110 for siSUMO1 and 146 for siSUMO2/3. Source data are provided as a Source Data file. **G** Western blot of chromatin fraction and WCL for myc, histone H3, SUMO2/3 and tubulin. Control and SUMO2/3 depleted U2OS cells were treated with 10 Gy of IR and collected for fractionation at indicated time points (performed once). Source data are provided as a Source Data file. **H** Western blot of chromatin fraction and WCL for myc, histone H3 and tubulin. U2OS cells expressing RNF168-WT, T208A or K210R mutants were treated with 10 Gy of IR and collected at indicated time points for fractionation (performed once). Source data are provided as a Source Data file.

## The RNF168-SPaCR motif is critical for IR resistance through limiting 53BP1

Given that both 53BP1 and BRCA1 are critical to DSB repair, we next investigated whether the PTM sites within the RNF168 SPaCR motif contribute to cellular IR resistance. We assessed the radiosensitivity of cells depleted of endogenous RNF168 and complemented with either WT or various SPaCR motif mutants using a colony survival assay. As expected, RNF168 depletion sensitized cells to IR. Cells remained sensitive when complemented with the T208A-RNF168 mutant, whereas resistance was restored by complementation with WT-RNF168 or with the T208A-P209A-RNF168 double mutant (Fig. 7A, C). These data are consistent with the ability of the P209A mutation to suppress the excessive chromatin accumulation of the T208A-RNF168 mutation.

We next investigated the importance of lysine-210 in regulating cellular resistance to IR. Notably, RNF168-depleted cells complemented with K210R-RNF168 or P209A-K210R-RNF168 variants remained sensitive to IR (Fig. 7B, C). These observations suggest that favouring trans-isomerization of RNF168 at P209 is insufficient to promote survival without K210 SUMOylation.

Finally, we aimed to test the cause of the radiosensitivity of cells expressing T208A-RNF168. We reasoned that this may arise because of altered or increased chromatin ubiquitination, increased 53BP1 presence or poor BRCA1 recruitment. To test the role of excessive 53BP1, we partially depleted 53BP1 in T208A-RNF168 complemented cells and assessed their ability to recruit BRCA1 to sites of DNA damage. Remarkably, 53BP1 reduction dramatically restored BRCA1 foci formation to levels similar to those seen in cells expressing WT-RNF168 (Fig. 7D, E). These data suggest that excessive 53BP1 is largely responsible for the poor BRCA1 recruitment observed in T208A-RNF168 expressing cells. To test the hypothesis that excessive 53BP1 accumulation on chromatin is detrimental to cells further, we partially depleted 53BP1 in T208A-RNF168 complemented cells and assessed their sensitivity to IR. In keeping with our hypothesis, partial depletion of 53BP1 restored the resistance of cells expressing the T208A RNF168 mutant to IR (Fig. 7F, G), supporting the notion that excessive 53BP1 accumulation is responsible for the increased IR- sensitivity of these cells.

Taking together, we propose a model in which RNF168 phosphorylation and PIN1-regulated isomerization promote its SUMOylation, ubiquitination and p97/VCP-mediated clearance from chromatin to prevent 53BP1-dependent suppression of BRCA1-recruitment to sites of DNA DSBs.

## Discussion

Restriction of RNF168-dependent Ub signaling is critical for balancing repair pathway choice at sites of DNA DSBs. Here, we show that limiting RNF168 chromatin accumulation requires a series of modifications of RNF168 itself within a newly defined regulatory motif we have called the SPaCR motif ('SUMO-PIN1-assisted Chromatin Regulator'). Within the SPaCR motif, threonine-208 is phosphorylated by CDK1/2, which promotes the binding of PIN1. PIN1 catalyzes isomerization of the threonine-208- proline-209 region, which promotes the SUMOylation and ubiquitination of RNF168 on lysine-210. Finally, the ubiquitination of RNF168 triggers p97/VCP-dependent extraction from chromatin. In the absence of this regulatory mechanism, excessive RNF168 on chromatin triggers aberrant 53BP1 accumulation that prevents BRCA1 recruitment to sites of DNA damage. Our data suggests that the toxic accumulation of 53BP1 compromises the repair capacity of cells.

Phosphorylation of threonine-208 within RNF168 requires CDK1/2 and is inhibited by the CDK1 inhibitor R03306. The cell cycle peak of pT208 is prior to mitosis, consistent with other CDK1 substrates. A mechanism that biases RNF168 clearance from chromatin in cells in late S-G2 would appear consistent with the need to bias repair away from excessive RNF168 and 53BP1 accumulation during the later parts of the cell cycle. We also observed increased WW-PIN1: RNF168 interaction (pT208 dependent) in asynchronous cells after IR, perhaps in common with several repair proteins modified by CDK1/2 after DNA damage (e.g., NBS1, CtIP, 53BP1, BRCA1/2)[67].

Proline isomerization can be a key regulatory step during the modification of proteins by ubiquitin and by extension, other small ubiquitin-like modifiers. PIN1-activity has been previously associated with both promoting substrate ubiquitylation, e.g., SF-1, CtIP, STK3 and CDK10[68–71], and preventing substrate ubiquitylation: e.g., p53 and BRD4[72–74]. Our observation that PIN1-dependent isomerization affects the capacity of local residues to be post-translationally modified by SUMO suggests that the RNF168 SPaCR might represent a previously undiscovered PIN1-regulated SUMO modification consensus motif that may exist in other PIN1 substrates. Interestingly, a mass spectrometry-based study from the Vertegaal group identifying SUMO and phosphorylation sites within the proteome revealed ninety-nine doubly SUMOylated and phosphorylated sequences, including six within a core SPaCR motif-type sequence "pS/TP**K**", Notably, all six of these proteins are known to bind chromatin (ZMYND8, TFAP4, BCLAF1, TCF20, YEATS2 and PRPF40A)[49]. However, further work is necessary to discern whether the SPaCR motif truly represents a bona fide regulatory motif. An alternative is that isomerization has an impact beyond the locality of proline-209 to allow SUMOylation or to reduce deSUMOylation. Moreover, while current models of PIN1 activity advocate isomerization of the proline local to the phosphorylation site, a model consistent with the observations in the current manuscript, current models also support the possibility of isomerization of more distal prolines[75]. Similarly, while our data shows that mutation of P209 → A, can overcome the deleterious impact of T208A, the rationale that a *trans*-T-A amide bond is the rescuing feature may be understating the mutation's impact. A limitation of our study is the inability to present a *cis*- T-P amide bond in cells to test the local role in a

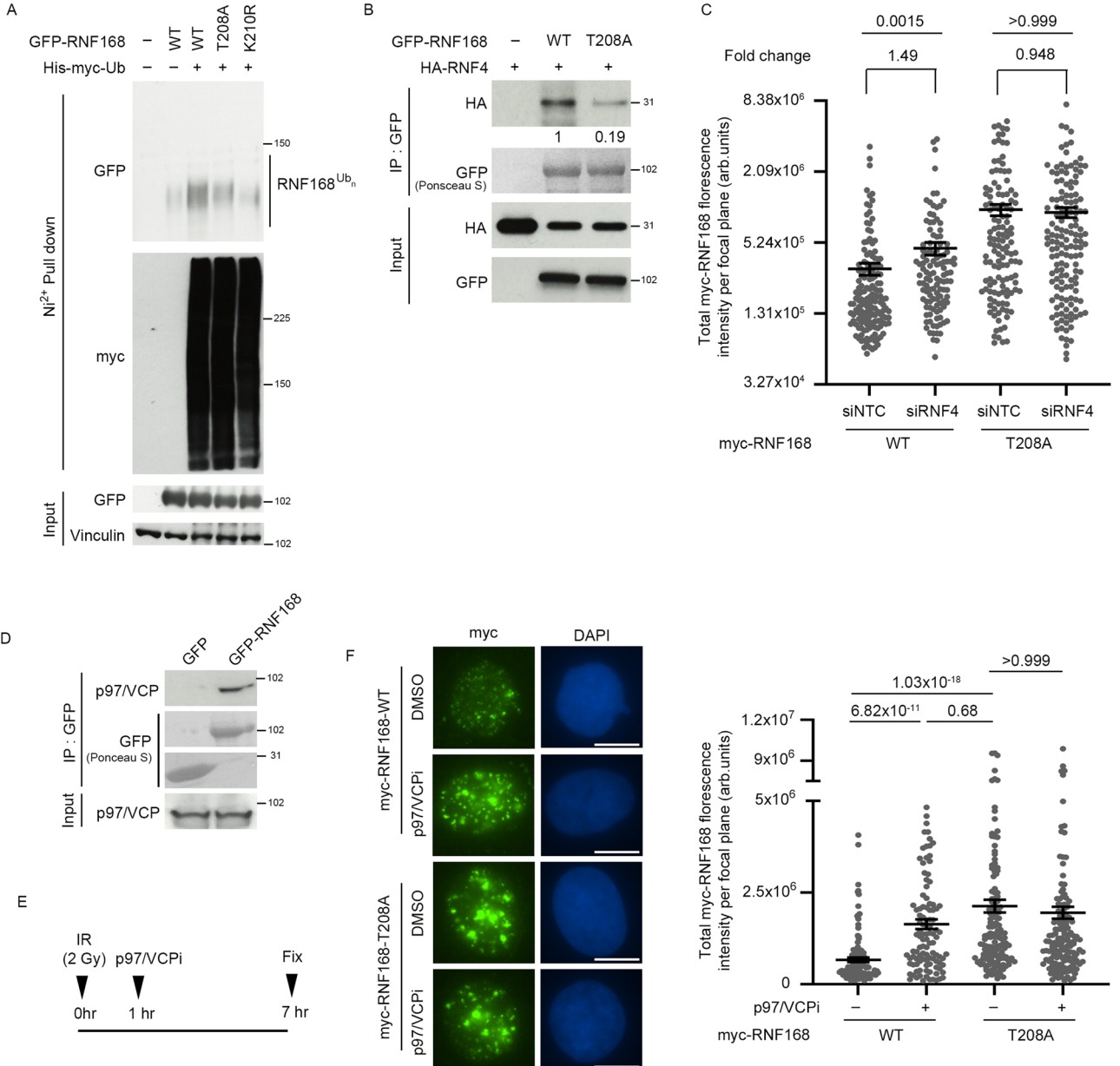

**Fig. 5 | T208A-RNF168 is resistant to RNF4 and p97/VCP-mediated suppression of accumulation. A** HEK293 cells transfected with GFP-WT-RNF168, GFP-T208A-RNF168 or GFP-K210R-RNF168 and His-myc-Ub. Ubiquitylated proteins were pulled down by His-Mag Sepharose Ni beads (Ni$^{2+}$ pull down) under denaturing conditions. Ubiquitylated RNF168 was detected by western blotting (Representative of 2 repeats). Source data are provided as a Source Data file. **B** HEK293 cells were transfected with GFP-WT-RNF168 or GFP-T208A-RNF168, along with HA-RNF4. GFP-Trap precipitation was performed, and precipitated proteins were analyzed using western blotting (Representative of 2 repeats). Source data are provided as a Source Data file. **C** Quantification of myc-RNF168 foci after radiation. U2OS cells expressing myc-RNF168-WT or myc-RNF168-T208A were treated with siNTC or siRNF4 and stained for myc, 1 hr post IR (2 Gy). myc intensity on *y*-axis. Data are mean ± s.e.m, *n* = 153 cells for myc-RNF168-WT+siNTC, 116 for myc-

RNF168-WT+siRNF4, 129 for myc-RNF168-T208A+siNTC and 148 for myc-RNF168-T208A+siRNF4. Source data are provided as a Source Data file. **D** HEK293 cells were transfected with GFP vector alone or with GFP-WT-RNF168. GFP-Trap precipitation was performed, and precipitated proteins were analyzed using western blotting (performed once). Source data are provided as a Source Data file. **E** Schematic of experimental design for (**F**). **F** p97/VCP inhibition results in hyperaccumulation of RNF168 after irradiation. U2OS cells expressing myc-RNF168-WT or T208A mutant were treated with 2 Gy IR and incubated an hour later with 1 μM CB-5083 for another 6 hrs. Cells were stained for myc and γ-H2AX. Scale bars 10 μm (Left). myc intensity on y-axis. Data is mean ± s.e.m, *n* = 126 cells for myc-RNF168 WT, 108 for myc-RNF168 WT+p97/VCPi, 152 for myc-RNF168 T208A and 159 for myc-RNF168 T208A + p97/VCPi. Source data are provided as a Source Data file.

physiological context or to measure the degree of *trans/cis* of that bond.

PIN1 has > 50 proposed substrates[74,75] including a growing number linked to repairing DNA damage. However, the impact of inhibiting PIN1 on different repair pathways is conflicting. For example, one reported that PIN1 suppressed CtIP activity, and consequently, PIN1-depleted cells exhibited reduced NHEJ and a slight increase in HR[69]. In

contrast, another study reported that PIN1 interacted with BRCA1 after damage and that loss of PIN1 activity reduced HR[65]. Our findings of epistasis between PIN1 and BRCA1 in IR-mediated cell sensitivity are consistent both with the phenotype of excessive RNF168-53BP1 and the regulation of BRCA1. Similarly, excessive 53BP1 has been shown to suppress NHEJ[25], so the poor repair of cells with excessive RNF168 may be attributable to alterations in more than one pathway. PIN1 is likely to

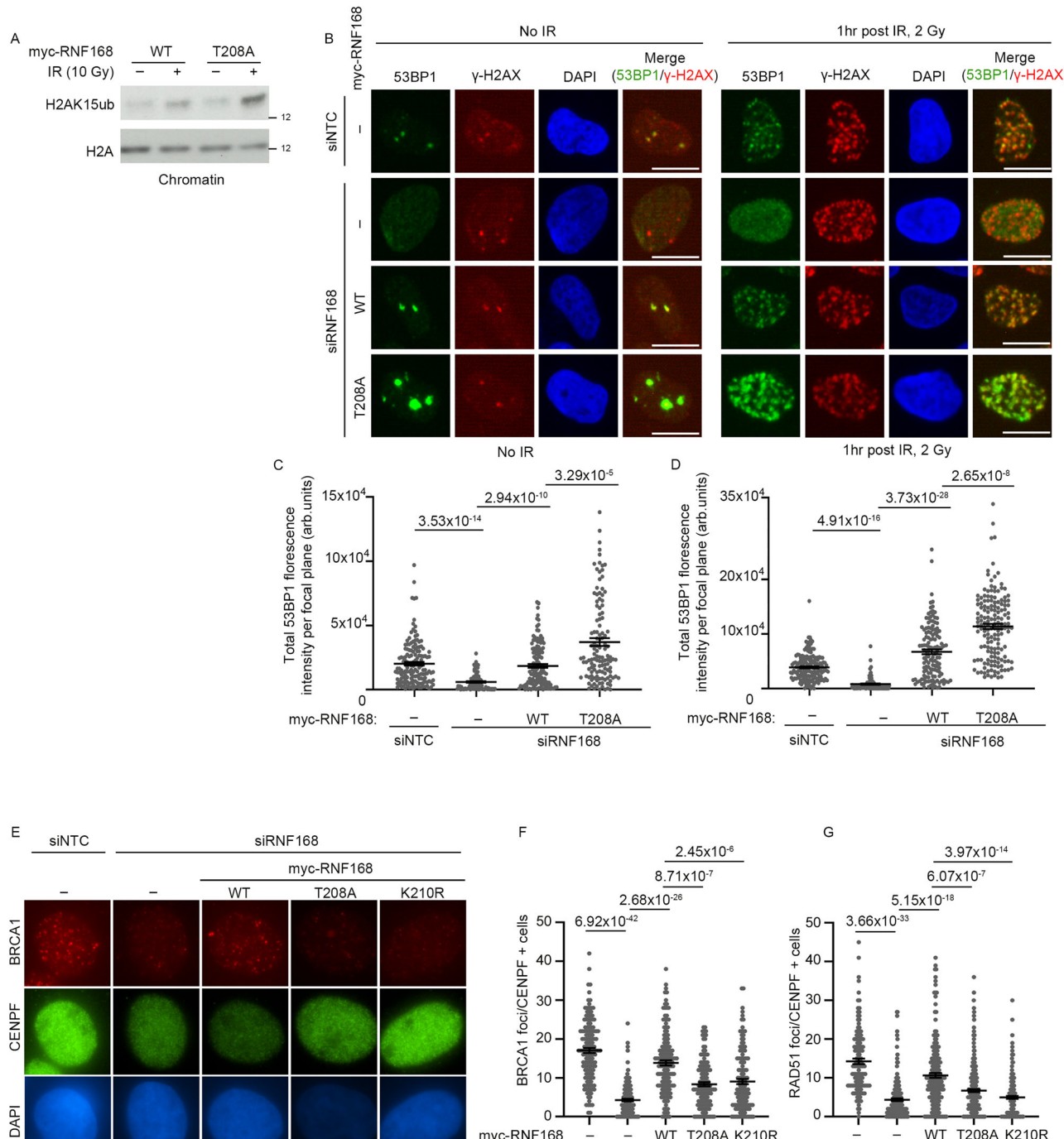

**Fig. 6 | T208A-RNF168 mutant drives increased chromatin ubiquitination, increased 53BP1 accumulation but reduced BRCA1 accumulation. A** Western blot of chromatin fraction for H2AK15ub and H2A from cells expressing siRNA-resistant myc-RNF168-WT or T208A. Endogenous RNF168 is depleted by siRNA. Cells were either left untreated or irradiated by 10 Gy IR and fractionated 1 hr later (Representative of 2 repeats). Source data are provided as a Source Data file. **B** U2OS cells treated with non-targeting siRNA or RNF168 siRNA and expressing siRNA-resistant myc-WT-RNF168 or myc-T208A-RNF168 were either untreated (left) or treated with 2 Gy of IR (right) and stained for 53BP1 and γH2AX. Scale bars 10 μm. **C** Quantification of 53BP1foci intensity from untreated cells. Data is mean ± s.e.m, $n = 150$ for siNTC, 100 for siRNF168, 140 for WT-RNF168 and 124 for T208A-RNF168. Source data are provided as a Source Data file. **D** Quantification of 53BP1 foci intensity from irradiated cells. Data is mean ± s.e.m, $n = 177$ for siNTC, 104 for

siRNF168, 133 for RNF168-WT and 158 for RNF168-T208A. Source data are provided as a Source Data file. **E** BRCA1 assessment in U2OS expressing siRNA resistant myc-RNF168-WT, T208A or K210R, depleted of endogenous RNF168 and treated with 2 Gy IR. 2 hrs post IR, cells were stained for BRCA1 and CENPF. Scale bars 10 μm. **F** Quantification of number BRCA1 foci from (**E**). Data is mean ± s.e.m, $n = 143$ cells for siNTC, 153 for siRNF168, 147 for RNF168-WT, 129 for RNF168-T208A and 136 for RNF168-K210R. Source data are provided as a Source Data file. **G** Quantification of number RAD51 foci in U2OS expressing siRNA-resistant myc-RNF168-WT, T208A or K210R, depleted of endogenous RNF168 and treated with 2 Gy IR. 2 hrs post IR, cells were stained for RAD51 and CENPF (representative images in Supplementary Fig. 6D). Data is mean ± s.e.m, $n = 132$ cells for siNTC, 207 for siRNF168, 195 for RNF168-WT, 297 for RNF168-T208A and 207 for RNF168-K210R. Source data are provided as a Source Data file.

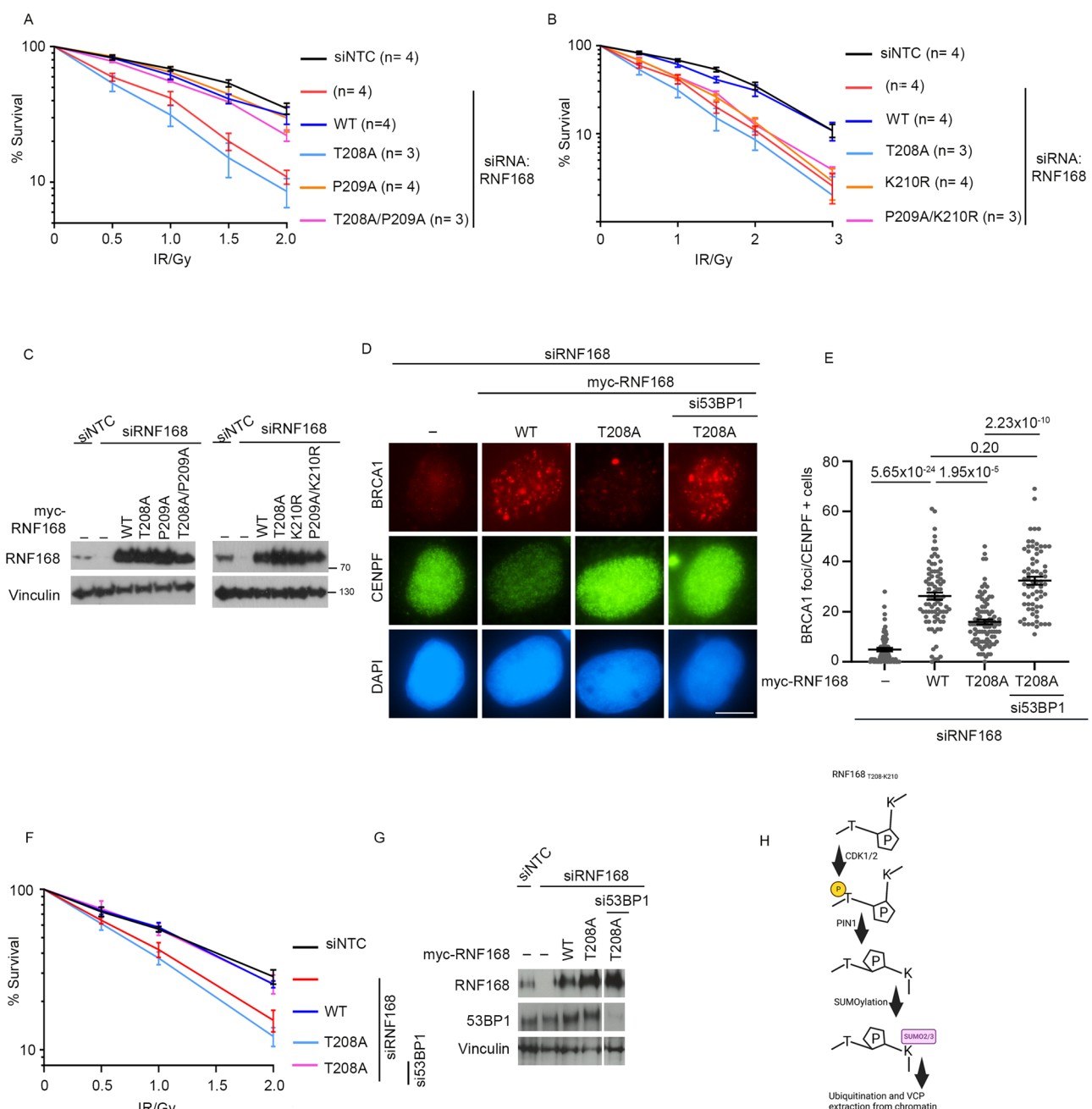

**Fig. 7 | The RINF168 SPaCR motif supports radio-resistance through suppression of 53BP1 accumulation. A** Colony survival of U2OS cells depleted of RNF168 and complemented with myc-WT-RNF168, T208A, P209A or T208A/P209A variants after treatment with indicated doses of IR. The number of replicates (n) for each condition is mentioned in parentheses. Data is mean ± s.e.m. p-value compared to siNTC at 2 Gy IR, $7.52 \times 10^{-7}$ for siRNF168, 0.91 for RNF168-WT, $4.7 \times 10^{-7}$ for RNF168-T208A, 0.73 for RNF168-P209A and 0.03 for RNF168-T208A/P209A. Source data are provided as a Source Data file. **B** Colony survival of U2OS cells depleted of RNF168 and complemented with myc-WT-RNF168, T208A, K210R and P209A/K210R variants of myc-RNF168 after treatment with indicated doses of IR. The number of replicates (n) for each condition is mentioned in parentheses. Data is mean ± s.e.m. p-value compared to siNTC at 2 Gy IR, $7.52 \times 10^{-7}$ for siRNF168, 0.91 for RNF168-WT, $4.7 \times 10^{-7}$ for RNF168-T208A, 0.00001 for RNF168-K210R and 0.00002 for RNF168-P209A/K210R. Source data are provided as a Source Data file. **C** Western blot to show depletion of RNF168 and complementation of RNF168 variants for (**A**) & (**B**) (performed

once). Source data are provided as a Source Data file. **D** BRCA1 assessment in U2OS expressing siRNA resistant myc-RNF168-WT or T208A, depleted of endogenous RNF168 and 53BP1 and treated with 2 Gy IR. 2 hrs post IR, cells were stained for BRCA1 and CENPF. Scale bars 10 μm. **E** Quantification of number BRCA1 foci from (**D**). Data is mean ± s.e.m, n = 77 cells for EV, 82 for myc-RNF168-WT, 92 for RNF168-T208A and 69 for RNF168-T208A+si53BP1. Source data are provided as a Source Data file. **F** Colony survival of U2OS cells depleted of RNF168 with or without si53BP1 and complemented with myc-WT-RNF168 or T208A after treatment with indicated doses of IR. n = 3. Data is mean ± s.e.m. p-value compared to siNTC at 2 Gy IR, 0.03 for siRNF168, 0.95 for RNF168-WT, 0.0058 for RNF168-T208A and 0.96 for RNF168-T208A + si53BP1. Source data are provided as a Source Data file. **G** Western blot to show depletion of RNF168, 53BP1 and complementation of RNF168 variants (performed once). Source data are provided as a Source Data file. **H** Illustration of the post-translational modifications at the RNF168 SPaCR 'SUMO-PIN1-assisted Chromatin Regulator' motif to regulate RNF168 dissociation from chromatin.

have a complex role in DNA repair, and many of its substrates, interactions and regulations are likely unknown.

Lysine-210 of RNF168 has recently been reported to be the main RNF168 SUMO acceptor site, modified through PIAS1, PIAS3 and PIASy SUMO E3 ligase activity[47]. Our mutation analysis suggested that modification at K210 is important for suppressing the spread of RNF168 on chromatin in a SUMO2/3- and RNF4-dependent manner. However, we cannot discount that disruption of lysine-210 may also compromise RNF168 ubiquitination directly. Since RNF4 suppression does not entirely phenocopy T208A or K210R mutations or p97/VCP inhibition, additional factors likely exist that contribute to RNF168 turn-over on chromatin. A recent report indicates that a fusion of SUMO to RNF168 results in phase-separated RNF168 puncta, which also contain 53BP1, suggesting the possibility that they sequester 53BP1[47]. Moreover, the SUMO-RNF168 fusion is a protein that fails to localize to regions of laser-induced DNA damage or to interact with chromatin[47], consistent with our observations that SUMOylation reduces chromatin association of RNF168.

Our data indicates that the SPaCR motif is an important determinant for cellular IR resistance. The ability of RNF168 to support PALB2-BRCA2-RAD51 recruitment[66] would appear to be insufficient to support cell survival, and instead, excessive RNF168 leads to toxic levels of 53BP1 accumulation. Our findings of excessive chromatin ubiquitination in the presence of T208A-RNF168 are similar to those observed when RNF168-ubiquitin signaling is disrupted by loss of deubiquitylating enzymes that restrict K13/15-Ub-H2A or K63-polyUb at chromatin. For example, loss of USP51[24], POH1[25] or USP11[27] and others[76], also result in increased 53BP1 recruitment to chromatin and radiosensitivity. In contrast, conditions of increased RNF168 protein levels associated with supra-physiological RNF168 spread around damage sites have been associated with radio-resistance[17,19]. Thus, increased chromatin ubiquitylation (caused by loss of a DUB, or failed RNF168 eviction) alters downstream protein recruitment in a manner that increased RNF168 protein levels alone do not. It may be relevant that Ub-chains are predicted to restrict the BARD1-Ub-nucleosome-core-particle interaction but not 53BP1-Ub-nucleosome-core-particle interaction[63,64]. A further unexplored area is the degree to which RNF169, which binds ubiquitinated H2A/H2AX and competes with 53BP1[28,29,77] is, or isn't, affected by the excessive histone ubiquitination resulting from RNF168 hyper-accumulation. Our data strongly suggest that the increased 53BP1 itself suppresses BRCA1 recruitment to the sites of DNA damage and increases radiosensitivity.

As PIN1, the SUMO conjugation system and p97/VCP are seen as potential therapeutic targets to improve the treatment of human cancers[78–80], the finding that they act in concert to limit RNF168 at chromatin adds to the view that suppressing PIN1[69], SUMOylation[53] or p97/VCP[81] may radically alter the cellular responses to DNA-damaging therapeutics. In summary, we find that the modification of the RNF168 SPaCR motif is part of the DNA-damage response, acting to promote RNF168 dissociation from damaged chromatin and promoting correct Ub-signaling and DNA repair.

## Methods
### Cell culture
Parental Flip-In U2OS, HEK293 and HeLa cells were cultured in DMEM media (Sigma, RNBK7590) supplemented with 10% free fetal bovine serum (FBS) (Gibco, 10500-064) and 1% penicillin and streptomycin. Hoechst DNA staining was regularly performed to test for mycoplasma.

### Cloning, site-directed mutagenesis and primers
myc-tagged wild type RNF168 cloned into pCDNA5/FRT/TO plasmid was obtained from JRM laboratory[54]. GFP-RNF168 plasmid generation is described in ref. 54. pGEX-5 × 3-GST-RNF168 vector was obtained from Prof. Grant Stewart's lab. Site-directed mutagenesis was

performed in-house by pfu DNA polymerase (Promega, M774A). Incorporated mutations were confirmed using Sanger sequencing (Source Biosciences Nottingham). N-terminal 6xHis-Flag-tagged SUMO2 was previously described[82]. All primers used for site-directed mutagenesis are listed in Supplementary Table 1.

### Generation of inducible stable cell lines
Stable cells were generated by transfecting parental Flip-In U2OS and HEK293 cells with pCDNA5/FRT/TO plasmid containing myc-RNF168 wild type or its mutants along with Flp recombinase cDNA containing pOG44 plasmid in 3:1 ratio. Control cells were seeded alongside and transfected with pOG44 plasmid alone. After 48 h, cells were selected by treatment with 100 μg/ml hygromycin (Thermo Fisher, 10687010). Expression of the gene of interest was induced by treating cells with 1 μg/ml doxycycline (Merck, D9891) for 72 h and confirmed by western blotting.

### Gene silencing and transfections
Transcript knockdown was performed by transfecting cells with siR-NAs. siRNA transfections were carried out with Dharmafect1 (Dharmacon, T-2001-03) as per manufacturer's instructions. The sequences of siRNAs used are listed in Supplementary Table 2. For inducible PIN1 knockdown, HEK293 cells were transfected with lentiviruses particles containing PIN1 shRNA (Sigma-Aldrich, TRCN0000010577) as per the manufacturer's instructions. After 24 h, stably transfected cells were selected by treatment with 2 μg/ml puromycin (Sigma, P7255). PIN1 knockdown was induced by adding 100 μM IPTG (Promega, V395A) for 48 h. Plasmid transfections were carried out by FuGENE 6 (Roche, E269A) as per the manufacturer's instructions.

### Ionizing radiation treatment
Cells were treated with ionizing radiation by CellRad Irradiator (Precision X Ray).

### Generation of GST-WW/W34A conjugated beads and Protein Expression
Glutathione S-transferases (GST)-tagged PIN1-WW domain conjugated beads were generated as described earlier[83]. Briefly, BL21 *E. coli* cells were transformed with pGEX protein expression vector containing cDNA for WW or W34A domain (obtained from JRM laboratory). Transformed colonies were later cultured in 50 ml LB media containing ampicillin for 16 h at 37°C. The next day, 5 ml of this starter culture was transferred to 500 ml fresh LB media containing ampicillin and cultured at 37°C till OD reached 0.6. At this stage, 1 mM IPTG was added to induce protein expression and cells were further cultured for 16 h at 16°C. Cells were then pelleted and lysed in 20 ml GST lysis buffer (20 mM Tris HCl pH 8, 130 mM NaCl, 1 mM EGTA, 1.5 mM MgCl2, 1% Triton X-100, 10% glycerol, 1 mM DTT) with the addition of two EDTA-free cOmplete protease inhibitor cocktail (Roche, 11836170001). Supernatant from lysed bacteria was collected by centrifugation at 16000 g for 10 min at 4°C. Cleared supernatants were then incubated with 500 μl glutathione sepharose 4B beads (GE Life Sciences, 17-0756-01) overnight at 4°C. Beads were washed three times with lysis buffer and resuspended in GST storage buffer (20 mM Tris HCl pH 8, 130 mM NaCl, 10% Glycerol and 1 mM DTT) at 50% volume.

N-terminal His$_6$-tag PIN1 plasmid was a gift from Dustin Maly (Addgene plasmid # 40773; http://n2t.net/addgene:40773; RRID:Addgene_40773). The PIN1 plasmid was transformed into Rosetta™ 2(DE3) pLysS Competent Cells (Millipore, lot #3517649) and PIN1 was expressed via auto-induction[84]. The culture was grown at 37 °C at 200 rpm until it reached an OD of 0.5, after which the temperature was dropped to 18 °C and left for 16 h. PIN1 was purified in a similar manner as previously reported[85] with the exception that PIN1 was lysed in a different buffer (50 mM Hepes, 300 mM NaCl, 1 mM TCEP and cOmplete mini (EDTA free) protease inhibitor cocktail at pH 7.4) and

elution for IMAC purification was conducted via a gradient from 0-500 mM imidazole. PIN1 tag was cleaved in a similar manner to Yang et al. 2014[86] except recombinant his-tagged Tobacco Etch Virus protease was used in place of thrombin. Tag-cleaved PIN1 was stored in 50 mM Hepes, 300 mM NaCl, 1 mM TCEP, 10% glycerol at pH 7.4 at -80 °C.

### PIN1-RNF168 peptide binding assay

Synthetic peptides of the RNF168 T208 motif were bought from GenScript Biotech (Supplementary Table 5). Lyophilized peptides were reconstituted in MilliQ water to a concentration of 1 mg/ml and then diluted in 100 mM ammonium acetate pH 6.8 to 25 µM. PIN1 was buffer exchanged into 100 mM ammonium acetate at pH 6.8 using 3,000 Da cut-off (Millipore) prior to native MS analysis. Buffer-exchanged PIN1 and a peptide were incubated together on ice at 2.5 µM and 12.5 µM, respectively. Protein-peptide complex formation was observed by native MS and the percentage bound of total protein was calculated using Eq. 1 below, in which the concentration of a complex at equilibrium can be defined by the sum of its peak intensities ($I$), when each peak is normalized to its charge[87] (Eq. 2). All native MS experiments were conducted on a QExactive HF Mass Spectrometer (Thermo Fisher Scientific) coupled with nanoelectrospray ionization using in-house pulled gold-coated borosilicate capillaries. Positive ionization mode was used throughout with the capillary voltage set to 1.4 kV. The source temperature was set at 250 °C, in source dissociation at 0, S-lens RF at 100. A mass range of 500–6000 m/z was set and acquired using a maximum ion injection time of 100 ms. The automatic gain control was set to $1 \times 10^6$ and with resolution set to 15,000. All raw data collected was analyzed in Xcalibur v4.1. All raw data can be found on the University of Birmingham data repository.

$$\% \, ligand \, bound \, to \, protein \, at \, equilibrium = 100 \left( \frac{[PL]_{eq}}{[P]_{eq} + [PL]_{eq}} \right) \quad (1)$$

$$[x]_{eq} = \sum_n (I(x^{n+})/n) \quad (2)$$

### $^1$H-$^1$H EXSY experiment

All NMR spectra were recorded at a calibrated temperature of 298 K on a 1-GHz Bruker Avance Neo spectrometer equipped with a 5-mm inverse HCN cryogenic probehead (helium-cooled) and running Topspin 4.1 acquisition software. Samples were prepared in a 90%:10% H2O:D2O mixture containing 50 mM NaCl, 50 mM Bis-Tris and 2 mM DTT at a pH of 6.6. The P209 peptide was dissolved to a final concentration of 2 mM. Where present, PIN1 was added to a final concentration of 25uM. EXSY spectra were recorded with a NOESY-based pulse-sequence[88], using a 3-9-19 WATERGATE element[89,90] for suppression of the water signal. The pseudo-3D EXSY experiment (multiple mixing-times) was recorded in an interleaved fashion, with the mixing-time loop inside the loop for the indirect dimension. The mixing-times for this pseudo-3D experiment were: 12.5, 25, 50, 75, 100, 150, 200, 250, 300, 350, 400, 500, and 600 ms.

NMR spectra were processed with NMRPipe v11.5[91] and visualized using CcpNmr AnalysisAssign[92]. The exchange rate-constant for the cis–trans isomerization was determined by extracting the intensity-ratio I_ct/I_tt from each of the 2D planes in the pseudo-3D EXSY spectrum and fitting the resulting time-dependent intensity-ratio profile to Eq. 3[43] using Levenburg-Marquardt least-squares minimization, as implemented in Python 3. Peak intensities were extracted as volumes using the lineshape-fitting software FuDA (https://www.ucl.ac.uk/hansen-lab/fuda/). Errors in the intensity-ratios were calculated from the estimated errors in the extracted peak volumes (as reported by FuDA) according to standard error propagation rules given in Eq. 4.

$$\frac{Ict}{Itt} = \frac{\{1 - \exp(-(k_{ct}+k_{tc})t_{mix})\}k_{tc}}{\{k_{ct}+k_{tc}\exp(-(k_{ct}+k_{tc})t_{mix})\}} \quad (3)$$

$$d\left(\frac{CT}{T}\right) = \sqrt{\left(\frac{dT}{T}\right)^2 + \left(\frac{dCT}{CT}\right)^2} \times \frac{CT}{T} \quad (4)$$

### Generation of Monoclonal antibody against pT208-RNF168

Mouse monoclonal antibody against phosphorylated Thr208 was generated by GenScript Biotech, (Netherlands) using standard protocol. Briefly, BALB/c mouse were immunized using synthetic peptide covering the sequence "SDPV(pThr)PKSEKKSKNC" of RNF168. Post immunization, polyclonal serum was collected, and antibody specificity was tested against phosphorylated and non-phosphorylated peptide using indirect ELISA. Specificity was also checked by western blotting of immunoprecipitated myc-WT-RNF168 and T208A full length proteins. After specificity was confirmed, hybridoma cells were generated and mAb were purified by protein A purification and specificity of mAb against pThr208 was again confirmed using indirect ELISA and western blotting as mentioned above.

### Pull-down assay

Cells were lysed in NP40 lysis buffer (50 mM Tris HCl pH 7.4, 250 mM NaCl, 5 mM EDTA, 50 mM, 1% Nonidet P40, pH 8.0) with the addition of cOmplete protease inhibitor cocktail (Roche, 11836170001) and PhosSTOP (Roche, 04906837001) at 4 °C. Cell debris was removed by centrifugation at 16000 g for 10 min at 4°C. Cell supernatant was then incubated with PBST (137 mM NaCl, 2.7 mM KCl, 8 mM Na2HPO4, 2 mM KH2PO4 and 1% Tween 20, pH 7.4) washed GST-WW or GST-W34A beads (20 µl) for 16 h at 4 °C. Beads were then washed three times with NP40 lysis buffer and boiled at 95 °C for 10 min in 30 µl 4x SDS loading buffer. Pulled down proteins were then run on SDS-PAGE and analyzed by western blotting.

### GFP-Trap affinity purification

GFP-RNF168 transfected HEK293T cells were lysed in NETN lysis buffer (150 mM NaCl, 50 mM Tris-Hcl pH 7.5, 2 mM MgCl2, 1% NP-40, 250U/ml Benzonase (Noagen), cOmplete protease inhibitor cocktail (Roche, 11836170001) and PhosSTOP (Roche, 04906837001)). 3–5 mg cleared lysate was incubated with 25 ul GFP-Trap agarose beads at 4 °C for 5 h. Later beads were washed three times with NETN buffer and boiled at 95 °C for 10 min in 30 µl 4x SDS loading buffer. Pulled down proteins were then run on SDS-PAGE and analyzed by western blotting.

### Denaturing nickel precipitation

HEK293 cells expressing 6xHis-Flag-SUMO2 or His-Myc-Ub were lysed in 8 M Urea buffer (8 M urea, 0.1 M Na2HPO4/NaH2PO4, 0.01 M Tris–HCl, pH 8, 10 mM β-mercaptoethanol) and sonicated at 10% intensity for 10 seconds. Cell debris was removed by centrifugation at 16000 g for 10 min at 4 °C. Cleared supernatant was incubated with 20 µl His Mag Sepharose Ni beads (Sigma, GE28-9673-88) overnight. Beads were washed three times with PBST. Precipitated proteins were eluted by boiling beads in 30 µl 4x SDS loading buffer and analyzed by western blotting.

### Cell cycle synchronization

Cells were synchronized at the G1/S boundary using a double thymidine block, as previously described[93]. Briefly, myc-RNF168 U2OS stable cells were treated with 2 mM thymidine for 20.5 h, released into fresh media for 7 h, and then treated with 2 mM thymidine for an additional 16.5 h. Following the second thymidine block, cells were released into fresh media for the indicated time points. Cells were harvested at these

time points, lysed, and analyzed by Western blotting to detect pT208-RNF168 and cell cycle markers.

## Western blotting

For western blotting, 50 μg cell lysates or pull-down samples were denatured by the addition of 4x SDS loading buffer and boiling at 95 °C for 10 min. Proteins were separated by their molecular weight by running onto 6%, 8%, and 12% SDS-PAGE gels and transferred onto PVDF membrane. Antibodies used for western blotting are listed in Supplementary Table 3.

## Preparation of chromatin fraction

U2OS cells were seeded at 50% confluency in 10 cm plates and treated with various siRNAs and 1 μg/ml doxycycline for 72 h. Cells were harvested by trypsinization and washed two times with PBS and resuspended in 500 μl PBS. 50 μl of cells were kept aside for the preparation of whole cell lysate (WCL). The remaining cells were resuspended in ice-cold sucrose buffer (10 mM Tris-Cl pH 7.5, 20 mM KCl, 250 mM Sucrose, 2.5 mM MgCl2, 0.3% Triton X-100, cOmplete protease inhibitor cocktail, PhosSTOP, 50 μM PR619 (Sigma, SML0430) and 20 μM MG132) and vortexed three times at low speed for 5 seconds. Cells were centrifuged at 500 g for 5 min at 4 °C and the supernatant was saved as cytoplasmic fraction. Pellet was resuspended in 200 μl NETN buffer (50 mM Tris-Cl pH 8.0, 150 mM NaCl, 2 mM EDTA, 0.5 % NP-40, cOmplete protease inhibitor cocktail, PhosSTOP, 50 μM PR619 and 20 μM MG132) and kept on ice for 30 min with intermittent tapping and centrifuged at 1700 g for 5 min at 4 °C. The supernatant was saved as the nuclear fraction and the remaining pellet (chromatin fraction) was resuspended in 200 μl NETN buffer. Both WCL and chromatin fraction were sonicated twice at 5% intensity for 10 seconds on ice and denatured by the addition of 4x SDS loading buffer and boiling at 95 °C for 5 min. Samples were analyzed by western blotting.

## Immunofluorescence microscopy and quantification of fluorescence intensity

For immunofluorescence microscopy, 1×10⁴ cells were plated on 13 mm glass coverslips. Cells were transfected with siRNAs and 1 μg/ml doxycycline for 72 h. For 53BP1, cell cultures were supplemented with 10 mM EdU for 30 min. After treatment with IR, cells were pre-extracted with 0.5% Triton X-100 in PBST for 5 min on ice and fixed with 4% paraformaldehyde. Cells were permeabilized by 0.5% Triton X-100 for 30 min at room temperature (RT). After blocking with 10% FCS, cells were incubated with primary antibodies for 1 hr, RT, except for mouse anti-myc mAb and rabbit anti-RNF168 polyclonal Ab (16 h, 4 °C). After three washes with PBST, cells were incubated with secondary antibodies for 1 hour, RT. DNA was stained with Hoechst stain (1:50000 dilution). Images were acquired using a Leica DM600B microscope with an HBO lamp with a 100-W mercury short arc UV-bulb light source and four filter cubes, A4, L5, N3 and Y5, to produce excitations at wavelengths of 360, 488, 555 and 647 nm, respectively. 100x oil immersion objective was used, and images were captured for each wavelength sequentially. Antibodies used for immunofluorescence are listed in Supplementary Table 3.

For fluorescence intensity analysis, images were analyzed using ImageJ software[94,95]. Region of interest (ROI) was drawn around the nucleus to calculate integrated density, area and mean fluorescence of background. Unlike irradiated cells, where the fluorescence intensity is analyzed from all the cells, fluorescence intensity from cells showing staining for γ-H2AX was analyzed in untreated cells. Corrected total cell fluorescence (CTCF) was calculated using the following formula: CTCF = Integrated Density−(Area of selected cell X Mean fluorescence of background readings). CTCF values from at least two independent experiments were plotted using GraphPad Prism9.

## Colony survival assay

Colony survival assays were performed as described previously[96]. 2 × 10⁴ U2OS cells were plated in 24 well cell plates. Cells were transfected with siRNAs and treated with 1 μg/ml doxycycline for 72 h. Cells were treated with indicated doses of IR before transferring them into six-well plate at limited density. Cells were cultured for 10–14 days. Colonies were stained with 0.5% crystal violet (BDH Chemicals) in 50% methanol and counted. Colony survival was calculated as the percentage change in colony formation following IR treatment compared to matched untreated cells. Each experiment is an average of three technical repeats, and the mean of three or more experiments was plotted using GraphPad Prism9.

## Statistics

Non-parametric Mann-Whitney test for used for data in Figs. 1B, E, G and S3F. Non-parametric Kruskal-Wallis test with Dunn's multiple comparison test was performed for data in Figs. 1I, J, 3D, E, 4F, 5C, F, 6C, D, F, G, 7E, S1D, S3B, and S4B. 2-way ANOVA was performed for colony survival assays. $p$-value < 0.05 was considered significant.

## Reporting summary

Further information on research design is available in the Nature Portfolio Reporting Summary linked to this article.

## Data availability

All data supporting the findings of this study are available within the paper and its Supplementary Information. Source data are available in Figshare repository at https://figshare.com/s/bcecb6a94f7d2eac289e.

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

## Acknowledgements

Wellcome Trust 206343/Z/17/Z (ASC, AJG, AKW, MJ) and CRUK C8820/A28283 (MM and HLM), and C17183/A23303 (GSS, SSJ); MRC: MR/W001152/1 (ASC) and MR/X001008/1(AJL). TC and JC are supported by a Leverhulme International Professorship award to TC (LIP-2020-017). We thank HWB-NMR at the University of Birmingham for providing access to high-field NMR instruments funded by the Wellcome Trust through the Biomedical Resources grant HWB-NMR: a national resource for biomolecular research (reference number: UNS43831). We thank the Microscopy and Imaging Services at Birmingham University (MISBU) in the Tech Hub facility for microscope support and maintenance and the Advanced Mass Spectrometry Facility at the University of Birmingham for support and maintenance of the mass spectrometers.

## Author contributions

A.S.C. generated stable cell lines, performed immunofluorescence experiments, pull-down assay, chromatin fractionation, western blotting, colony survival assay and analyzed data. M.J.W.M. performed protein purification and performed native mass spectrometry analysis. J.C.

performed $^{1}$H-$^{1}$H EXchange SpectroscopY experiments. A.J.L. performed colony survival for p97/VCPi and BRCA1/PIN1 co-depletion experiments. A.J.G. and M.J. performed cloning and site-directed mutagenesis. A.S.C. and A.K.W. performed SUMO IP. A.S.C., H.L.M. carried out cell cycle synchronization. S.S.J. assisted in chromatin fractionation experiment. J.R.M. and A.S.C. wrote the paper. A.L. supervised M.J.W.M., G.S.S. supervised A.S.C. and T.C. supervised J.C. J.R.M. supervised A.S.C., M.J.W.M., A.J.L., M.J., H.L.M., A.J.G., A.K.W. A.S.C., A.J.G. and J.R.M. conceptualized. All authors commented on the paper.

## Competing interests

The authors declare no competing interests.
