## [Transparent Peer Review file · Nature Communications]

PIN1-SUMO2/3 motif suppresses excessive RNF168 chromatin accumulation and ubiquitin signalling to promote IR resistance.

Corresponding Author: Professor Joanna Morris

Version 0:

Reviewer comments:

Reviewer #1

(Remarks to the Author)

In this manuscript, Morris and colleagues identify the role of PIN1, a peptidyl-prolyl isomerase, in promoting double strand DNA (DSB) repair via controlling SUMO-dependent RNF168 turnover at sites of DNA damage induced by IR. Prompted by the observation that PIN1 deficiency leads to accumulation of RNF168 on chromatin, the authors delineated the signaling cascade of posttranslational modifications that promote PIN1 and SUMO-mediated RNF168 clearance from IR-induced damaged sites to facilitate DSB repair. Specifically, they identified that phosphorylation of Thr208 of RNF168 by CDK1/2 allows the recognition of RNF168 by PIN1, which in turn promotes RNF168 SUMOylation at an adjacent residue Lys210, resulting in p97/VCP-dependent extraction of RNF168 from chromatin. This regulated turnover of RNF168 was shown to be critical for limiting the supraphysiological accumulation of RNF168, thereby promoting homology-directed repair and conferring IR resistance.

Overall, this is an interesting story that reveals a new role of PIN1 in DSB repair process. Experiments are well-controlled, and most of the structure-function analyses are convincing. There are extensive analyses of all the players involved in RNF168 modification and stability, which is a major strength of the manuscript. Nevertheless, the molecular mechanism through which the catalytic activity of PIN1 drives the SUMOylation of RNF168 is lacking, and at the same time, it is not clear whether prolyl isomerization of RNF168 is indeed the cause of subsequent SUMO modification. Below are specific concerns regarding these points, which need to be addressed further to strengthen the manuscript.

1. There is currently little evidence to support that PIN1-dependent conformational changes drive RNF168 SUMOylation. For instance, both RNF168 T208A (phosphorylation-dead) and K201R (SUMO-dead) mutants show similar phenotypes, but a defect in phosphorylation rather than prolyl isomerization may be responsible for a defect in subsequent SUMOylation. As PIN1-dependent SUMO modification is the key point of the manuscript, more direct evidence is needed to conclude that prolyl isomerization of RNF168 is prerequisite for SUMO modification.
2. Even if prolyl isomerization of RNF168 should lead to SUMO modification, the underlying mechanism is currently lacking. It could be a conformational change that promotes the interaction with a SUMO conjugating enzyme or it may interfere with deSUMOylation activity. It is expected that RNF168 may adapt a distinct structural change via prolyl isomerization, but exactly how it will impact downstream SUMO activity needs to be addressed.
3. Based the inhibitory study using RO-3306, the authors conclude that CDK1 is involved in the phosphorylation of RNF168. CDK1 primarily controls the G2/M transition and is cell-cycle regulated. On the other hand, PIN1-mediated SUMO regulation appears to be dependent on DNA damage (i.e., IR, which damages DNA independently of cell cycles) and also to be constitutively on. The impact of cell cycle in RNF168 modification and its implication to DSB repair needs to be addressed.
4. Additionally, there are other well-known kinases responsible for the phosphorylation of PIN1 substrates. These need to be tested to further substantiate the major role of CDK1/2 responsible for RNF168 modification.
5. One informative mutant is RNF168 P209A/K209R, which is trans-prone but SUMO-defective, and has been shown to be loss-of-function. The chromatin retention of this mutant needs to be checked in comparison to P209A to further confirm that it

is indeed refractory to p97 due to a defect in SUMO modification although the PIN1 requirement is bypassed.

6. In many cases, total fluorescence was plotted in +/- IR separately. The value between no IR and after IR appears not much different, while it is expected to have a damage-dependent effect. Analysis of data with combined sets may be helpful to clarify the RNF168 dynamics on chromatin before and after IR damage.

7. There are obvious mistakes where many sentences are duplicated (e.g., page 3 line 82; page 3 line 99; page 6 line 173), which need to be corrected.

Reviewer #2

(Remarks to the Author)

In the manuscript "PIN1-SUMO2/3 mediated RNF168 turn-over promotes IR resistance", Morris and colleagues describe a regulatory mechanism that restricts Rnf168 activity during response to DNA damage induced by ionizing radiation (IR). The authors present data that Rnf168 is recognized by Pin1, likely in a Cdk1-dependent manner. Pin1 binding is proposed to promote cis-trans isomerization of Rnf168, leading to its modification by Sumo2/3, ubiquitination and Vcp-mediated extraction of Rnf168 from chromatin. This pathway is suggested to limit ubiquitin signaling at DNA DSBs, balance recruitment of 53bp1 vs Brca1 to DNA damage sites and promote radioresistance.

This study provides mechanistic insight into the DNA damage response pathways that can be of interest to the readership of Nature Communications. However, I feel that the manuscript should be significantly strengthened to support the current model and would strongly benefit from additional experiments that clarify the mechanism or explain its functional relevance. In particular, the authors should provide more convincing sumoylation assays, correctly assess Rnf168-dependent ubiquitination at the DSB sites and perform further experiments to document the role of Cdk1/2 and Vcp in the proposed mechanism.

To enhance the dataset, the authors could provide in vitro data for Pin1- and Cdk1-dependent modification of Rnf168, explore how the proposed mechanism relates to previously established regulation of Rnf168 by the Trip12/Ubr5 ligases or how it is coordinated with other known Pin1 functions in DNA repair. Addressing these and/or other points, listed below would make the manuscript much more interesting to a broader readership.

1. General comments to the authors model:

a) How is the proposed mechanism coordinated with previously described ubiquitin pathways that limit Rnf168 activity at DSBs, such as that mediated by Trip12/Ubr5? Some experiments (e.g., Figure 4h) indicate that Pin1 largely mediates basal extraction of Rnf168 from chromatin and that Pin1-resistant variants are efficiently removed from chromatin following release from IR. Quantification of such assays may help estimate the contribution of Pin1.

b) This study would greatly benefit from any evidence that Pin1 actually isomerizes Rnf168 (or a relevant peptide) in cells or in vitro (e.g., using the isomer-specific proteolysis assay).

c) Even for proline, the fraction of the cis isomer usually does not exceed 10%. Is this compatible with the authors' model?

d) Is it possible to predict, based on known Pin1 structures, whether Pin1 binding at pT208 would be compatible with sumoylation of K210? Since the Pin1 binding site is immediately adjacent to the proposed Sumo acceptor, Pin1 may sterically hinder access to the lysine side chain when bound to Rnf168. The authors could also analyze whether Pin1 binding is diminished for the cis isomer, which could facilitate sumoylation of K210.

e) The data suggest that Cdk1, which is thought to primarily act in G2/M, is the main kinase that targets Rnf168 at T208 (as the specific Cdk1 inhibitor RO-3306 abolishes T208 phosphorylation). How does this affect the proposed model - e.g., is the mechanism restricted to a subset of DNA repair events that occur late in the cell cycle?

2. A previous report showed that Pin1 is required for Brca1-dependent HR (Luo et al, 2020). This can account for / contribute to the radiosensitivity phenotype of Pin1-depleted cells and should be discussed. Another study, cited in the manuscript (ref 55), found that Pin1 deficiency increases HR, opposite to the argument of the present work. A different recent report found that sumoylation of Rnf168 promotes its phase separation and antagonizes its function at DSB sites (Wei et al, 2023). The authors should discuss their data in the context of these published studies and, ideally, perform experiments that address how these distinct functions of Pin1 may be coordinated.

3. While the Rnf168-Pin1 interaction is analyzed in some detail, connection to Cdk kinases and the Vcp complex is only superficially assessed by chemical inhibitors. This should be strengthened using additional assays (e.g., using sh/sgRNA, loss-of-function alleles and in vitro kinase reactions).

4. Interactions of Rnf168 with Pin1, Cdk1/2 and Vcp should be analyzed by immunoprecipitation or PLA assays with endogenous proteins, as suitable antibodies are commercially available.

5. The authors show that loss of Pin1-dependent regulation of Rnf168 leads to radiosensitivity due to ectopic 53bp1 recruitment. One might expect that Brca1-deficient cells would be highly sensitive to inhibition of Pin1 in combination with IR. Is this the case? Authors could also determine the impact of Cdk1/2 or Vcp inhibition in combination with IR on radioresistance to strengthen their conclusions and highlight possible clinical implications of the study.

6. It's not clear why MG132 is used for the assays that assess Rnf168 sumoylation, as the authors' argument is that Sumo2/3 conjugation doesn't affect Rnf168 stability (line 152). Instead, experiments shown in Figures 4g and 5a,b should use a general cysteine proteinase inhibitor, such as N-Ethylmaleimide, and include IR treatment as a positive control.
7. Sumo2 conjugates on Rnf168 in Figures 4g and 5b are hardly distinguishable. As an alternative assay, authors could use denaturing IP with Rnf168 antibodies followed by immunoblotting with Sumo2/3 antibodies. His-Sumo pulldown assays could be performed with fractionated lysates to increase the proportion of modified Rnf168. Here the authors could also test whether Sumo/Ub-modified Rnf168 accumulates in the nuclear fraction upon inhibition of Vcp, in support of their model.
8. The labeling of Ub-pH2AX bands in Figure 6a is not justified - at best, these can be any (combinations of) mono- or di-Ub pH2AX, including diUb on K118, K119 or K127. Since Rnf168 selectively modifies K13/15, authors should use a specific commercially available antibody to K13/15-ubiquitinated H2A to test the impact of Rnf168 mutations. Along these lines, it is not clear which bands are quantified in Figure 6b.
9. K210 is both sumoylated and ubiquitinated (phosphosite.org) and the K210R mutation should directly reduce Rnf168 ubiquitination. Can the authors distinguish this possibility from effects of Sumo-modified K210 experimentally? For example, can the authors identify sites targeted by a Sumo-directed ubiquitin ligase, which recognizes sumoylated K210? Experiments with knockdown of candidate Sumo ligases would also support the proposed requirement for sumoylation in Rnf168 extraction from chromatin.
10. The authors use different IR treatment protocols for immunofluorescence assays (2Gy) and for immunoblots and pulldown assays (10Gy). At least some key assays should be performed under the same conditions to validate the results.
11. The cell lysate fractionation experiments should show a soluble fraction. Quantification of these assays would also strengthen the conclusions.
12. The total levels of Rnf168 in Figure 5c should be shown.
13. Authors use siRNA in cell survival assays which take around two weeks. At this time after transfection, the siRNA is most likely lost, which compromises the experiment. These assays should use stable depletion using constitutive or tet-inducible shRNA; alternatively authors could validate the knockdown at two weeks after siRNA transfection.
14. The statistical test used to derive p-values is most likely inappropriate.
Fig. 1B,E,G; 6B,C; S2F: For using Welch's t-test, the assumption of normality needs to be maintained. However, the distribution of data in most graphs indicates that the assumption of normality is violated and it is not mentioned whether the authors used a test to confirm the opposite. For comparison of two populations with unequal variances and without normal distribution, the non-parametric Mann-Whitney test can be used.
Fig. 1I,J; 3B,E,F; 4B,C; 5F; 6F,G; 7B,D: As in a), the assumption of normality needs to be maintained for applying Welch's t-test. Additionally, no corrections for multiple testing were performed. For comparison of more than two groups in one experiment, an appropriate and non-parametric test is the Kruskal-Wallis test followed by a suitable post-hoc test such as Dunn's multiple comparison.
15. Fig. 2E and Fig. 7 E,G,I: Do the survival curves significantly differ from each other?
16. The authors should proofread and edit the manuscript, as there are many errors. For example, sentences on lines 81-86, 99-105 and 173-175 are repeated; ref 21 is cited twice (line 61); there are typos in lines 263 and 316-318; the focal plane is called "plan" in axis labels of the immunofluorescence quantification graphs, etc. The assay referred to as "colony survival assay" is usually called colony formation assay or cell survival assay.

Reviewer #3

(Remarks to the Author)

In this study by Chauhan et al., the authors delineate a molecular mechanism that suppresses excessive and uncontrolled RNF168-driven ubiquitin signaling at damaged chromatin. Their compelling and elegant results reveal the importance of three adjacent residues in RNF168 (Thr208, Pro209, and Lys210) that act as a switch to trigger the chromatin extraction of RNF168 by the ubiquitin-selective segregase VCP/p97.

The elegant cell biology and biochemical approaches reveal key steps in the underlying mechanism. Initially, Thr208 is phosphorylated by CDK1/2. This triggers the binding and cis/trans isomerization of Pro209 by the peptidyl-prolyl isomerase PIN1. Mutating Pro209 to Ala209 already promotes trans isomerization, alleviating the need for Thr208 phosphorylation and PIN1 activity. Lys210 is subsequently SUMOylated and ubiquitylated in a manner that depends on both Thr208 phosphorylation and PIN1 activity. The residues Thr208 and Lys210 are required to limit H2A ubiquitylation and 53BP1 binding to damaged chromatin, and they are important in providing resistance to ionizing radiation-induced DNA double-strand breaks.

The paper is well-written, and all the experiments are convincing and support the conclusions. All methods and reagents are clearly described and organized in tables. The level of molecular detail is extraordinary, and this work provides an important piece of the puzzle, thereby advancing the DNA damage response and ubiquitin/SUMO fields.

I support the publication of this manuscript in Nature Communications, provided the authors appropriately address the following points:

MAJOR POINTS

- (1) The current title "PIN1-SUMO2/3-Mediated RNF168 Turnover Promotes IR Resistance" may not fully capture the content of the manuscript, I believe. The authors could consider: "A novel PIN1/SUMO regulatory motif governs RNF168 removal to suppress excessive ubiquitin signaling and promote DNA damage tolerance", or something else.
- (2) Overexpression of RNF168 tends to increase its aggregation, potentially increasing the need to regulate its chromatin-bound levels. While Fig 1A clearly demonstrates increased RNF168 fluorescence for endogenous RNF168 after PIN1 knockdown, indicating that this principle applies to endogenous RNF168, most other panels rely on myc-RNF168 as a readout. It is worth noting that myc-RNF168 is expressed at considerably higher levels than endogenous RNF168, as depicted in Fig 2F. Therefore, it is important to provide evidence of increased chromatin retention through western blot analysis, similar to what is shown in Fig 1H, for endogenous RNF168 following either PIN1 depletion or the use of PIN1 inhibitors
- (3) While it is subjective, the acronym CARPS (Chromatin Association Regulated by PIN1-SUMO) may not be very catchy. The authors could consider using another acronym, such as SPARC (SUMO/PIN1-Assisted Removal from Chromatin), SPECS (SUMO/PIN1 Extraction from Chromatin to Safeguard), or something else.
- (4) The authors demonstrate that CDK1/2-mediated phosphorylation of Thr208 is the switch that leads to VCP/p97-mediated extraction from chromatin. Using the phospho-Thr208 antibody generated in this study, have the authors observed differences in this phosphorylation event during the cell cycle, particularly when CDK1/2 activity is lower? To what extent does the cell cycle contribute to the phosphorylation-triggered extraction of RNF168?
- (5) Results: "Furthermore, consistent with this observation, these two RNF168 mutants also failed to support RAD51 foci formation to the level observed with WT-RNF168, consistent with their impact on BRCA1"

The lack of BRCA1 foci rescue by RNF168-T208 is somewhat surprising, considering that BARD1 binds H2A-K15-Ub (PMID: 34321663), which is increased in RNF168-T208 cells. Does the RNF168-T208A/P209A double mutant rescue this phenotype, considering that this mutant still supports VCP-mediated extraction from chromatin? Can the BRCA1 phenotype in RNF168-T208 cells be rescued by knockdown of 53BP1?

RNF168 also binds PALB2 and stimulates RAD51 recruitment through its C-terminal PID region (PMID: 28240985). Although this goes beyond the scope of the current manuscript, it could be mentioned that this function of RNF168 might be affected by the T208/K210 mutations, potentially contributing to the RAD51 defect observed in Fig 7C. While not essential here, it would be interesting to test if the RNF168-T208/K210 mutants still interact with PALB2.

MINOR POINTS

- (i) Perhaps it is better to switch Fig 1D/E and Fig 1F/G to follow the description in the results section.
- (ii) The γ H2AX intensity also seems increased in the image shown in Fig 3D for the RNF168-T208A mutant. Has this been quantified along with myc-RNF168 levels? If not different, it might be better to select another image here.
- (iii) The detection of SUMO2-conjugated myc-RNF168 in the example shown in Fig 2G is not very convincing (mainly due to the non-specific band). Is there not another more convincing example the authors could use here, more similar to the experiment shown in Fig 5A?
- (iv) For consistency, the authors should decide on 1-letter or 3-letter annotation for amino acids. Now both are used in the text and figures.
- (v) VCP/p97 is used 6 times, while p97/VCP is used 14 times. Choose one and use consistently. Or introduce as VCP/p97 once and use either VCP or p97 throughout the manuscript.

TEXTUAL

- (a) "RNF168 modified H2A is bound by 53BP1" in introduction. Hyphen between RNF168-modified.
- (b) "RNF168 contains two Ub binding domains" in introduction. Hyphen between Ub-binding domains.
- (c) "in turn driving p97/VCP mediated clearance" in the introduction. Hyphen between p97/VCP-mediated clearance.
- (d) The following sentence in the results is duplicated: "We considered that direct post-translational modification (PTM) of RNF168 could represent one possible mechanism to regulate its activity and mined publicly available "eukaryotic linear motif (ELM) resource" 30 for potential RNF168 regulatory motifs."

(e) "numerous potential PIN1 binding sites near", should be PIN1-binding sites (also in Fig S1A, and elsewhere in the manuscript).

(f) The following sentence in the results is duplicated: "In addition, we also observed excessive RNF168 chromatin accumulation both in untreated and IR-treated cells incubated with small molecule inhibitors of PIN1 that either affect its enzymatic activity (Juglone 31 and PiB 32) or its stability (ATRA 33) (Figure 1I & J, Supplementary Figure 1C)."

(g) Please check the legend on the y-axis in Fig 1I/J, Fig 3E/F and Fig 5F: "intensity per focal plan". I assume focal plane is meant here?

(h) Results: "used these to test purification of RNF168 from cell extracts" rephrase to clarify. For instance, used these to co-immunoprecipitate RNF168 from cell extracts.

(i) Results: "We observed increased PIN1 and RNF168204-210 mass products .." annotate RNF168204-210 differently for clarity. Perhaps 204-2010-RNF168, or RNF168 (204-2010)?

(j) The following sentence in the results is duplicated: "The modification of RNF168 by SUMO1 is known to be mediated by PIAS4, and has been implicated in promoting its transcription 45."

(k) Results: "we depleted SUMO1 and SUMO2 with SUMO3". Rephrase to clarify. Perhaps we depleted SUMO1 and SUMO2/3?

(l) Discussion: "clearance of RNF168 appearsn to influence", correct to "appears".

Version 1:

Reviewer comments:

Reviewer #1

(Remarks to the Author)

The authors addressed most of previous concerns. Although the effort to identify the molecular mechanism through which PIN1-induced conformational changes impact subsequent SUMO modification did not turn out positive, the manuscript was overall much improved.

Reviewer #2

(Remarks to the Author)

In the revised version of the manuscript "PIN1-SUMO2/3 motif suppresses excessive RNF168 chromatin accumulation and ubiquitin signalling to promote IR resistance.", Morris and coworkers thoroughly addressed the concerns raised in the original review, both experimentally and in the text. The dataset now provides a much more solid support for the authors' model. I believe that this work will be of interest to researchers in the areas of DNA damage response and posttranslational protein regulation.

Reviewer #3

(Remarks to the Author)

The authors have carefully addressed all my points and have significantly improved an already excellent manuscript. I have no reservations about supporting the publication of this manuscript in Nature Communications.

Response to reviewer's comments NCOMMS-23-40431

We thank the reviewers for taking the time to review our manuscript and for providing insightful comments to improve it further. Below, we provide a point-by-point response to the comments raised.

Reviewer #1:

In this manuscript, Morris and colleagues identify the role of PIN1, a peptidyl-prolyl isomerase, in promoting double strand DNA (DSB) repair via controlling SUMO-dependent RNF168 turn-over at sites of DNA damage induced by IR. Prompted by the observation that PIN1 deficiency leads to accumulation of RNF168 on chromatin, the authors delineated the signaling cascade of posttranslational modifications that promote PIN1 and SUMO-mediated RNF168 clearance from IR-induced damaged sites to facilitate DSB repair. Specifically, they identified that phosphorylation of Thr208 of RNF168 by CDK1/2 allows the recognition of RNF168 by PIN1, which in turn promotes RNF168 SUMOylation at an adjacent residue Lys210, resulting in p97/VCP-dependent extraction of RNF168 from chromatin. This regulated turn-over of RNF168 was shown to be critical for limiting the supraphysiological accumulation of RNF168, thereby promoting homology-directed repair and conferring IR resistance.

Overall, this is an interesting story that reveals a new role of PIN1 in DSB repair process. Experiments are well-controlled, and most of the structure-function analyses are convincing. There are extensive analyses of all the players involved in RNF168 modification and stability, which is a major strength of the manuscript. Nevertheless, the molecular mechanism through which the catalytic activity of PIN1 drives the SUMOylation of RNF168 is lacking, and at the same time, it is not clear whether prolyl isomerisation of RNF168 is indeed the cause of subsequent SUMO modification. Below are specific concerns regarding these points, which need to be addressed further to strengthen the manuscript.

1. There is currently little evidence to support that PIN1-dependent conformational changes drive RNF168 SUMOylation. For instance, both RNF168 T208A (phosphorylation-dead) and K201R (SUMO-dead) mutants show similar phenotypes, but a defect in phosphorylation rather than prolyl isomerisation may be responsible for a defect in subsequent SUMOylation. As PIN1-dependent SUMO modification is the key point of the manuscript, more direct evidence is needed to conclude that prolyl isomerisation of RNF168 is prerequisite for SUMO modification.

2. Even if prolyl isomerisation of RNF168 should lead to SUMO modification, the underlying mechanism is currently lacking. It could be a conformational change that promotes the interaction with a SUMO conjugating enzyme or it may interfere with deSUMOylation activity. It is expected that RNF168 may adapt a distinct structural change via prolyl isomerisation, but exactly how it will impact downstream SUMO activity needs to be addressed.

We have shown that RNF168 mutants, T208A (phosphorylation-dead) and K201R (SUMO-dead), show similar phenotypes with respect to the association of RNF168 on chromatin. This strongly suggests that the phosphorylation and SUMOylation of RNF168 are linked. Several observations point to PIN1 mediating this link. Our observation that the double-mutant T208-P209A can restore normal chromatin accumulation and SUMOylation of RNF168 T208A single mutant and can also rescue the increased IR sensitivity of cells expressing this RNF168 T208A mutant (Figure 3B-E, 4C, 7A) strengthens our hypothesis that PIN1-dependent isomerisation of P209 is critical for regulating RNF168 turnover on chromatin. Further, in vitro evidence is discussed below.

As the reviewer suggested, we have expanded our mechanistic studies of how PIN1 regulates RNF168 function:

1. Using NMR, we now show that PIN1 isomerises the SDPV(pT)PK motif of RNF168 (Fig 3A).

2. We then tested whether the confirmation of P209 can directly alter local K210 SUMOylation. To address this, we synthesised an RNF168 peptide bearing a 5,5-dimethyl proline (Dmp) in place of proline-209 and replaced all the lysines, except K210, within this peptide with arginines. The non-natural 5,5-dimethylproline (Dmp) constrains the T-Pro amide bond into a *cis*-conformation (DOI: 10.1111/j.1399-3011.2005.00257.x [1]). We also synthesised a similar RNF168 peptide in which the proline-209 was replaced with alanine to bias towards a *trans* confirmation of the T-A amide. These peptides were then tested for their relative ability to be SUMOylated using Ubc9 and recombinant SUMO and being deSUMOylated by the SENP1 catalytic region. Unfortunately, after spending months trying to optimise this in vitro SUMOylation reaction, the results from these experiments were inconclusive and we feel their presentation would be inappropriate.

We can discuss the reason for this (peptide Vs full-length protein, the absence of E3 ligase in the assays, etc). Certainly, we don't feel as though this reaction properly represents the situation in cells. It is unlikely that we will be able to get the detailed mechanistic insight about the PIN1-dependent SUMOylation of RNF168 that the reviewer requires. Our discussion reinforces the statement that we make no claims regarding the underpinning mechanism linking PIN1 activity and SUMOylation.

Nevertheless, we have presented ample amounts of data demonstrating that RNF168 phosphorylation/SUMOylation is essential for regulating its association with chromatin. Furthermore, our demonstration that depletion of PIN1 mimics the cellular phenotypes caused by mutating RNF168, T208 and K210 clearly indicates that PIN1 plays a critical role in linking the post-translational modification of these sites, which we believe is sufficient for this publication.

3. Based the inhibitory study using RO-3306, the authors conclude that CDK1 is involved in the phosphorylation of RNF168. CDK1 primarily controls the G2/M transition and is cell-cycle regulated. On the other hand, PIN1-mediated SUMO regulation appears to be dependent on DNA damage (i.e., IR, which damages DNA independently of cell cycles) and also to be constitutively on. The impact of cell cycle in RNF168 modification and its implication to DSB repair needs to be addressed.

We have confirmed that T208 phosphorylation depends on CDK1/2 through kinase depletion (Figure 2K) and additionally shown that RNF168 interacts with CDK1/2 (Supplementary Figure 2C). As suggested, we also addressed cell cycle regulation, noting that pT208 levels peak 4-8 hours after release from the thymidine block and before the appearance of pSer10-histone-3, a timing consistent with late-S-G2 (Supplementary Figure 2D). Thus, the pattern of pT208 matches that expected for a CDK1/2 substrate. We discuss the role of CDK1/2 in the damage response in the discussion (we are not the first to see this).

4. Additionally, there are other well-known kinases responsible for the phosphorylation of PIN1 substrates. These need to be tested to further substantiate the major role of CDK1/2 responsible for RNF168 modification.

As suggested by the reviewer, we screened a panel of proline-directed Ser/Thr kinase inhibitors to identify the potential kinases responsible for T208-RNF168 phosphorylation. Our results indicate that CDK1/2 is the primary kinase targeting T208-RNF168 (Supplementary Figure 2B). We further validated these findings through siRNA-mediated depletion experiments, where the depletion of CDK1/2 nearly eliminates the pT208 signal (Figure 2K). Therefore, if there are other kinases that target T208, we don't feel that they make a significant contribution to regulating RNF168.

5. One informative mutant is RNF168 P209A/K209R, which is trans-prone but SUMO-defective, and has been shown to be loss-of-function. The chromatin retention of this mutant needs to be checked in comparison to P209A to further confirm that it is indeed refractory to p97 due to a defect in SUMO modification although the PIN1 requirement is bypassed.

We thank the reviewer for this suggestion and have completed the experiment (Supplementary Figure 4C), which demonstrates that, unlike the T208A mutant, the P209A mutant cannot rescue the excessive chromatin recruitment of the K210R RNF168 mutant.

6. In many cases, total fluorescence was plotted in $-/+$ IR separately. The value between no IR and after IR appears not much different, while it is expected to have a damage-dependent effect. Analysis of data with combined sets may be helpful to clarify the RNF168 dynamics on chromatin before and after IR damage.

We apologize for this. To clarify, we have quantified the recruitment of RNF168 in untreated cells showing signs of endogenous damage (identified by γ -H2AX staining). However, these are only a small proportion of the total population of cells. We have modified the materials and methods section to explain our analysis methods.

7. There are obvious mistakes where many sentences are duplicated (e.g., page 3 line 82; page 3 line 99; page 6 line 173), which need to be corrected.

We apologise for these mistakes. We have now removed the duplicate sections from our revised manuscript.

Reviewer #2:

In the manuscript "PIN1-SUMO2/3 mediated RNF168 turn-over promotes IR resistance", Morris and colleagues describe a regulatory mechanism that restricts Rnf168 activity during response to DNA damage induced by ionising radiation (IR). The authors present data that Rnf168 is recognised by Pin1, likely in a Cdk1-dependent manner. Pin1 binding is proposed to promote cis-trans isomerisation of Rnf168, leading to its modification by Sumo2/3, ubiquitination and Vcp-mediated extraction of Rnf168 from chromatin. This pathway is suggested to limit ubiquitin signaling at DNA DSBs, balance recruitment of 53bp1 vs Brca1 to DNA damage sites and promote radioresistance.

This study provides mechanistic insight into the DNA damage response pathways that can be of interest to the readership of Nature Communications. However, I feel that the manuscript should be significantly strengthened to support the current model and would strongly benefit from additional experiments that clarify the mechanism or explain its functional relevance. In particular, the authors should provide more convincing sumoylation assays, correctly assess Rnf168-dependent ubiquitination at the DSB sites and perform further experiments to document the role of Cdk1/2 and Vcp in the proposed mechanism. To enhance the dataset, the authors could provide in vitro data for Pin1- and Cdk1-dependent modification of Rnf168, explore how the proposed mechanism relates to previously established regulation of Rnf168 by the Trip12/Ubr5 ligases or how it is coordinated with other known Pin1 functions in DNA repair. Addressing these and/or other points, listed below would make the manuscript much more interesting to a broader readership.

1. General comments to the authors model:

a) *How is the proposed mechanism coordinated with previously described ubiquitin pathways that limit Rnf168 activity at DSBs, such as that mediated by Trip12/Ubr5? Some experiments (e.g., Figure 4h) indicate that Pin1 largely mediates basal extraction of Rnf168 from chromatin and that Pin1-resistant variants are efficiently removed from chromatin following release from IR. Quantification of such assays may help estimate the contribution of Pin1.*

We have tested TRIP12/UBR5 loss as suggested. Interestingly, we found an additive impact of depleting TRIP12/UBR5 on the chromatin association of RNF168 in cells expressing the T208 mutant (Supplementary Figure 5A). These findings indicate that the PIN1- and TRIP12/UBR5-dependent pathways act independently to regulate RNF168.

b) *This study would greatly benefit from any evidence that Pin1 actually isomerises Rnf168 (or a relevant peptide) in cells or in vitro (e.g., using the isomer-specific proteolysis assay).*

We have addressed this aspect and shown, as anticipated, that PIN1 isomerises the SDPV(pT)PK RNF168 site (Figure 3A).

c) *Even for proline, the fraction of the cis isomer usually does not exceed 10%. Is this compatible with the authors' model?*

This is a good point, but it is hard to say what the proportion of *cis/trans* RNF168 protein is. In the 2012 paper from the Kun Ping Lu lab (Nakamura et al.), 9% of phospho-tau peptides were in the *cis*-isoform. However, using an isomer-specific antibody, Nakamura et al. saw an almost equal proportion of *cis* and *trans* forms of cellular tau (PMCID: 3601591 [2]). Therefore, all we can conclude from our data is that the *trans* form is protective against excessive chromatin accumulation. However, we have discussed the limitations of our study with respect to the relative levels of *cis* versus *trans* isoforms of endogenous RNF168 in the manuscript.

d) *Is it possible to predict, based on known Pin1 structures, whether Pin1 binding at pT208 would be compatible with sumoylation of K210? Since the Pin1 binding site is immediately adjacent to the proposed Sumo acceptor, Pin1 may sterically hinder access to the lysine side*

chain when bound to Rnf168. The authors could also analyse whether Pin1 binding is diminished for the *cis* isomer, which could facilitate sumoylation of K210.

We tested the reviewer's idea directly by assessing whether co-incubation of PIN1 with a phosphorylated RNF168 peptide might impact SUMOylation. We made wild-type and catalytically inactive PIN1 (C113S) and examined the impact on peptide SUMOylation. The addition of the C113S PIN1 mutant in the SUMOylation reactions moderately reduced peptide SUMOylation, although this did not reach statistical significance (Reviewer Figure 1). The inclusion of WT PIN1 in the SUMOylation reaction did not increase peptide SUMOylation levels. From this, we can infer that either the Ubc9-dependent SUMOylation reaction occurs quicker than the isomerisation reaction or that in the context of a peptide rather than an entire protein, isomerisation has little impact on the ability of the peptide to be modified by Ubc9. We suspect that the reduction in peptide SUMOylation in the presence of the catalytically inactive PIN1 probably results from the C113S mutant binding the phospho-T208 residue and blocking any subsequent modifications of the peptide since it is unable to catalyse the proline isomerisation. We should also point out that this *in vitro* SUMOylation reaction does not require an E3 ligase, so it is likely that this reaction does not completely recapitulate the situation in cells.

As outlined in our response to a similar comment made by reviewer 1, we also carried out *in vitro* SUMOylation assays using a phospho-peptide of RNF168 containing pT208 and K210, where P209 was either replaced by an alanine residue or a 5,5-dimethyl modified proline (Dmp) to mimic the *cis* isomer. As indicated in our response to reviewer 1, we spent a large amount of trying to optimise this SUMOylation reaction using recombinant Ubc9 and SUMO. However, the results from this assay were inconclusive. Thus, whilst we are unable to specifically demonstrate that the isomerisation of RNF168 proline-209 by PIN1 enhances its SUMOylation *in vitro*, we believe that we have presented a large amount of supporting data in cells to indicate the phosphorylation of T208 by CDK1/2 promotes the recruitment of PIN1, which isomerises P209 to stimulate the SUMOylation of K210.

e) The data suggest that Cdk1, which is thought to primarily act in G2/M, is the main kinase that targets Rnf168 at T208 (as the specific Cdk1 inhibitor RO-3306 abolishes T208 phosphorylation). How does this affect the proposed model - e.g., is the mechanism restricted to a subset of DNA repair events that occur late in the cell cycle?

We have confirmed that T208 phosphorylation depends on CDK1/2 through kinase depletion (Figure 2K) and additionally shown that RNF168 interacts with CDK1/2 (Supplementary Figure 2C). As suggested, we also addressed cell cycle regulation, noting that pT208 levels peak 4-8 hours after release from the thymidine block and before the appearance of pSer10-histone-3, a timing consistent with late-S/G2 (Supplementary Figure 2D). Thus, the pattern of pT208 matches that expected for a CDK1/2 substrate. Our observation that a CDK1-resistant RNF168 mutant, T208A, suppresses RAD51 foci in CENPF-positive (largely G2) cells (Figure 6E-F), is consistent with the notion that RNF168 eviction would be primed in S-phase/G2.

2. A previous report showed that Pin1 is required for Brca1-dependent HR (Luo et al, 2020). This can account for / contribute to the radiosensitivity phenotype of Pin1-depleted cells and

should be discussed. Another study, cited in the manuscript (ref 55), found that Pin1 deficiency increases HR, opposite to the argument of the present work. A different recent report found that sumoylation of Rnf168 promotes its phase separation and antagonises its function at DSB sites (Wei et al, 2023). The authors should discuss their data in the context of these published studies and, ideally, perform experiments that address how these distinct functions of Pin1 may be coordinated.

Many thanks for these points. The role of PIN1 in the DNA damage response is complex and the published literature is somewhat contradictory. We have expanded our discussion to reflect this. Additionally, to address the reviewer's question pertaining to the relationship between PIN1 and BRCA1, we investigated whether loss of PIN1 and BRCA1 had epistatic or additive effects on cellular sensitivity to IR. In this context, we found an epistatic relationship between PIN1 depletion and BRCA1 loss, consistent with the Luo *et al* paper (Supplementary Figure 6E). Importantly, this result is consistent with our finding that the reduced recruitment BRCA1 to sites of DNA damage in T208A-RNF168 complemented cells could be restored by 53BP1 deletion (Figure 7E). Therefore, we propose a model whereby high levels of RNF168 chromatin accumulation drives excessive 53BP1 accumulation, which in turn, suppresses BRCA1 recruitment and HR-dependent DNA DSB repair.

3. While the Rnf168-Pin1 interaction is analysed in some detail, connection to Cdk kinases and the Vcp complex is only superficially assessed by chemical inhibitors. This should be strengthened using additional assays (e.g., using sh/sgRNA, loss-of-function alleles and in vitro kinase reactions).

We have strengthened both aspects. We have shown that RNF168 interacts with both CDK1/2 and p97/VCP and strengthened our findings using CDK1/2 and p97/VCP inhibitors with siRNA knockdown of these enzymes (Figure 2K, Supplemental Figure 2C, Figure 5D).

4. Interactions of Rnf168 with Pin1, Cdk1/2 and Vcp should be analysed by immunoprecipitation or PLA assays with endogenous proteins, as suitable antibodies are commercially available.

The co-immunoprecipitation studies of RNF168 with CDK1/2 and p97/VCP have been carried out with endogenous proteins (Supplemental Figure 2C, Figure 5D).

5. The authors show that loss of Pin1-dependent regulation of Rnf168 leads to radiosensitivity due to ectopic 53bp1 recruitment. One might expect that Brca1-deficient cells would be highly sensitive to inhibition of Pin1 in combination with IR. Is this the case? Authors could also determine the impact of Cdk1/2 or Vcp inhibition in combination with IR on radioresistance to strengthen their conclusions and highlight possible clinical implications of the study.

We have shown that loss of PIN1 and BRCA1 are epistatic with respect to regulating cellular sensitivity to IR (Supplementary Fig 6E). On reflection, this might have been expected given the poor BRCA1 recruitment in cells complemented with T208A RNF168 mutant. Additionally, we have also demonstrated that treating cells with the p97/VCP inhibitor after IR exposure increases their sensitivity to IR (Supplementary Figure 5D).

6. It's not clear why MG132 is used for the assays that assess Rnf168 sumoylation, as the authors' argument is that Sumo2/3 conjugation doesn't affect Rnf168 stability (line 152). Instead, experiments shown in Figures 4g and 5a,b should use a general cysteine proteinase inhibitor, such as N-Ethylmaleimide, and include IR treatment as a positive control.

We have repeated the experiments without MG132 and observed the same effect, i.e. loss of PIN1 or mutation of K210 reduces RNF168 SUMOylation (Figure 4A&B).

7. Sumo2 conjugates on Rnf168 in Figures 4g and 5b are hardly distinguishable. As an alternative assay, authors could use denaturing IP with Rnf168 antibodies followed by immunoblotting with Sumo2/3 antibodies. His-Sumo pulldown assays could be performed with fractionated lysates to increase the proportion of modified Rnf168. Here the authors could also test whether Sumo/Ub-modified Rnf168 accumulates in the nuclear fraction upon inhibition of Vcp, in support of their model.

We have repeated these experiments as suggested and observed an improved level of RNF168 SUMOylation (Figure 4 A & B, Figure 5A). Additionally, we observed that the treatment of cells with the p97/VCP inhibitor increases the level of chromatin-bound RNF168 (Figure 5 E & F).

8. The labeling of Ub-pH2AX bands in Figure 6a is not justified - at best, these can be any (combinations of) mono- or di-Ub pH2AX, including diUb on K118, K119 or K127. Since Rnf168 selectively modifies K13/15, authors should use a specific commercially available antibody to K13/15-ubiquitinated H2A to test the impact of Rnf168 mutations. Along these lines, it is not clear which bands are quantified in Figure 6b.

We agree with the reviewer's comments. As suggested, we have now included data using an antibody specific for K13/15-ubiquitinated H2A (Figure 6A).

9. K210 is both sumoylated and ubiquitinated (phosphosite.org) and the K210R mutation should directly reduce Rnf168 ubiquitination. Can the authors distinguish this possibility from effects of Sumo-modified K210 experimentally? For example, can the authors identify sites targeted by a Sumo-directed ubiquitin ligase, which recognises sumoylated K210? Experiments with knockdown of candidate Sumo ligases would also support the proposed requirement for sumoylation in Rnf168 extraction from chromatin.

The knockdown of SUMO2/3 supports the role of SUMO in mediating RNF168 extraction. Indeed we cannot rule out that K210 may also be a site of ubiquitination; indeed, our data shows that mutation of K210R reduces the ubiquitination of RNF168. Despite this, in our revised manuscript, we have demonstrated an interaction between RNF168 and the STUbL RNF4, which is disrupted by the T208A mutation (Figure 5B). Additionally, we have shown that RNF4 depletion leads to increased RNF168 chromatin localization (Figure 5C, Supplementary Figure 5B), suggesting a potential role for RNF4 in RNF168 turn-over. These data are consistent with RNF168 chromatin turnover being dependent on a SUMO- and ubiquitin-dependent pathway. However, we have revised the discussion of our revised manuscript to include the possibility that SUMOylation or ubiquitination of K210 are both potential mechanisms with which the cell regulates the chromatin association of RNF168.

10. The authors use different IR treatment protocols for immunofluorescence assays (2Gy) and for immunoblots and pulldown assays (10Gy). At least some key assays should be performed under the same conditions to validate the results.

We have now examined RNF168 localization at both 2Gy and 10Gy IR in immunofluorescence experiments (Supplementary Figure 1D). In addition, it is worth mentioning that when assessing the chromatin association of RNF168, we also see more RNF168 accumulation on chromatin in unirradiated cells following PIN1 or SUMO2/3 depletion (Figure 1H, Figure 4H), or when T208A/K210R RNF168 mutants are expressed (Figure 4I).

11. The cell lysate fractionation experiments should show a soluble fraction. Quantification of these assays would also strengthen the conclusions.

Now shown, Figure 1C. Quantification is also included in the figures.

12. The total levels of Rnf168 in Figure 5c should be shown.

Input shown (now Figure 5A).

13. Authors use siRNA in cell survival assays which take around two weeks. At this time after transfection, the siRNA is most likely lost, which compromises the experiment. These assays should use stable depletion using constitutive or tet-inducible shRNA; alternatively authors could validate the knockdown at two weeks after siRNA transfection.

Survival assays performed as we have undertaken them are optimised to have depletion at the time of IR exposure and acute recovery. The grow-time for the colonies allows a quantification of the surviving fraction after damage and recovery. Our experiments are controlled (control siRNA, RNF168 add-backs etc), and thus, we are confident that this approach is reasonable to address the questions at hand. We note that depletion and add-backs using colony survival assays is routine in many other publications to answer similar questions e.g. Nature Comms papers: Richards et al., 2023, PMID: PMC10354178 [6], Kelliher et al., 2024, PMID: PMC11143218 [7], and Sci Ad Kuster et al., 2021 PMID: PMC7895427 [8].

14. The statistical test used to derive p-values is most likely inappropriate.

Fig. 1B,E,G; 6B,C; S2F: For using Welch's t-test, the assumption of normality needs to be maintained. However, the distribution of data in most graphs indicates that the assumption of normality is violated and it is not mentioned whether the authors used a test to confirm the opposite. For comparison of two populations with unequal variances and without normal distribution, the non-parametric Mann-Whitney test can be used. Fig. 1I,J; 3B,E,F; 4B,C; 5F; 6F,G; 7B,D: As in a), the assumption of normality needs to be maintained for applying Welch's t-test. Additionally, no corrections for multiple testing were performed. For comparison of more than two groups in one experiment, an appropriate and non-parametric test is the Kruskal-Wallis test followed by a suitable post-hoc test such as Dunn's multiple comparison.

We thank the reviewer for guiding us in the right statistical methods to use. We have re-analysed the data, and significant values are included for each panel. Specifically, we have used non-parametric Mann-Whitney test for Figure 1B, 1E, 1G, S3F, non-parametric Kruskal-Wallis test with Dunn's multiple comparison test for Figure 1I, 1J, 3D, 3E, 4F, 5C, 5F, 6C, 6D, 6F, 6G, 7E, S1D, S3B, S4B and 2-Way ANOVA for Figure 2E, 7A, 7B, 7F, S5C, S6E. The statistical methods used are described in the material and methods section.

15. Fig. 2E and Fig. 7 E,G,I: Do the survival curves significantly differ from each other? Significance values are now mentioned in the figures and explained in the material and methods section. A 2-Way ANOVA test is used for statistical significance for Figures 2E, 7A, 7B, 7F, S5C, S6E.

16. The authors should proofread and edit the manuscript, as there are many errors. For example, sentences on lines 81-86, 99-105 and 173-175 are repeated; ref 21 is cited twice (line 61); there are typos in lines 263 and 316-318; the focal plane is called "plan" in axis labels of the immunofluorescence quantification graphs, etc. The assay referred to as "colony survival assay" is usually called colony formation assay or cell survival assay. Our sincere apologies – these are now corrected.

Reviewer #3 (Remarks to the Author):

In this study by Chauhan et al., the authors delineate a molecular mechanism that suppresses excessive and uncontrolled RNF168-driven ubiquitin signaling at damaged chromatin. Their compelling and elegant results reveal the importance of three adjacent residues in RNF168 (Thr208, Pro209, and Lys210) that act as a switch to trigger the chromatin extraction of RNF168 by the ubiquitin-selective segregase VCP/p97.

The elegant cell biology and biochemical approaches reveal key steps in the underlying mechanism. Initially, Thr208 is phosphorylated by CDK1/2. This triggers the binding and cis/trans isomerisation of Pro209 by the peptidyl-prolyl isomerase PIN1. Mutating Pro209 to Ala209 already promotes trans isomerisation, alleviating the need for Thr208 phosphorylation and PIN1 activity. Lys210 is subsequently SUMOylated and ubiquitylated in a manner that depends on both Thr208 phosphorylation and PIN1 activity. The residues Thr208 and Lys210 are required to limit H2A ubiquitylation and 53BP1 binding to damaged chromatin, and they are important in providing resistance to ionising radiation-induced DNA double-strand breaks.

The paper is well-written, and all the experiments are convincing and support the conclusions. All methods and reagents are clearly described and organised in tables. The level of molecular detail is extraordinary, and this work provides an important piece of the puzzle, thereby advancing the DNA damage response and ubiquitin/SUMO fields.

I support the publication of this manuscript in Nature Communications, provided the authors appropriately address the following points:

MAJOR POINTS

(1) The current title "PIN1-SUMO2/3-Mediated RNF168 Turn-over Promotes IR Resistance" may not fully capture the content of the manuscript, I believe. The authors could consider: "A novel PIN1/SUMO regulatory motif governs RNF168 removal to suppress excessive ubiquitin signalling and promote DNA damage tolerance", or something else.

We have expanded the title to "PIN1-SUMO2/3 motif suppresses excessive RNF168 chromatin accumulation and ubiquitin signalling to promote IR resistance".

(2) Overexpression of RNF168 tends to increase its aggregation, potentially increasing the need to regulate its chromatin-bound levels. While Fig 1A clearly demonstrates increased RNF168 fluorescence for endogenous RNF168 after PIN1 knockdown, indicating that this principle applies to endogenous RNF168, most other panels rely on myc-RNF168 as a readout. It is worth noting that myc-RNF168 is expressed at considerably higher levels than endogenous RNF168, as depicted in Fig 2F. Therefore, it is important to provide evidence of increased chromatin retention through western blot analysis, similar to what is shown in Fig 1H, for endogenous RNF168 following either PIN1 depletion or the use of PIN1 inhibitors
Agreed, undertaken in Figure 1C.

(3) While it is subjective, the acronym CARPS (Chromatin Association Regulated by PIN1-SUMO) may not be very catchy. The authors could consider using another acronym, such as SPARC (SUMO/PIN1-Assisted Removal from Chromatin), SPECS (SUMO/PIN1 Extraction from Chromatin to Safeguard), or something else.
These are much better than our original name! We've borrowed these ideas to create "SUMO-PIN1-assisted Chromatin Regulator" (SPaCR). Thanks.

(4) The authors demonstrate that CDK1/2-mediated phosphorylation of Thr208 is the switch that leads to VCP/p97-mediated extraction from chromatin. Using the phospho-Thr208 antibody generated in this study, have the authors observed differences in this

phosphorylation event during the cell cycle, particularly when CDK1/2 activity is lower? To what extent does the cell cycle contribute to the phosphorylation-triggered extraction of RNF168?

As suggested, we also addressed cell cycle regulation, noting that pT208 levels peak 4-8 hours after release from the thymidine block and before the appearance of pSer10-histone-3, a timing consistent with late-S-G2 (Supplementary Figure 2D). Thus, the pattern of pT208 matches that expected for a CDK1/2 substrate.

(5) Results: "Furthermore, consistent with this observation, these two RNF168 mutants also failed to support RAD51 foci formation to the level observed with WT-RNF168, consistent with their impact on BRCA1"

The lack of BRCA1 foci rescue by RNF168-T208 is somewhat surprising, considering that BARD1 binds H2A-K15-Ub (PMID: 34321663), which is increased in RNF168-T208 cells. Does the RNF168-T208A/P209A double mutant rescue this phenotype, considering that this mutant still supports VCP-mediated extraction from chromatin? Can the BRCA1 phenotype in RNF168-T208 cells be rescued by knockdown of 53BP1?

We were also surprised and for the same reasons. As suggested, we addressed whether the BRCA1 phenotype is rescued by the knockdown of 53BP1, and we found that indeed it is (Figure 7D & E). Thus, the excessive 53BP1 recruitment is clearly the defect. Presumably, 53BP1 is favoured over BRCA1 by the excessive Ub, and we speculate how this may arise in the discussion.

RNF168 also binds PALB2 and stimulates RAD51 recruitment through its C-terminal PID region (PMID: 28240985). Although this goes beyond the scope of the current manuscript, it could be mentioned that this function of RNF168 might be affected by the T208/K210 mutations, potentially contributing to the RAD51 defect observed in Fig 7C. While not essential here, it would be interesting to test if the RNF168-T208/K210 mutants still interact with PALB2. As the loss of 53BP1 rescues the defects of T208A-mutant (Figure 7D, E & F), the increased RNF168:PALB2:RAD51, if affected, may not be too deleterious, we have mentioned the interaction, as requested.

MINOR POINTS

(i) Perhaps it is better to switch Fig 1D/E and Fig 1F/G to follow the description in the results section.

Now changed.

(ii) The γ H2AX intensity also seems increased in the image shown in Fig 3D for the RNF168-T208A mutant. Has this been quantified along with myc-RNF168 levels? If not different, it might be better to selected another image here.

Apologies – a more representative image is shown (Figure 3B & C).

(iii) The detection of SUMO2-conjugated myc-RNF168 in the example shown in Fig 4G is not very convincing (mainly due to the non-specific band). Is there not another more convincing example the authors could use here, more similar to the experiment shown in Fig 5A?

Repeated, shown in Figure 4A and Figure 4B

(iv) For consistency, the authors should decide on 1-letter or 3-letter annotation for amino acids. Now both are used in the text and figures.

Revised.

(v) VCP/p97 is used 6 times, while p97/VCP is used 14 times. Choose one and use consistently. Or introduce as VCP/p97 once and use either VCP or p97 throughout the manuscript.

Revised.

TEXTUAL

(a) “RNF168 modified H2A is bound by 53BP1” in introduction. Hyphen between RNF168-modified.

Revised.

(b) “RNF168 contains two Ub binding domains” in introduction. Hyphen between Ub-binding domains.

Revised.

(c) “in turn driving p97/VCP mediated clearance” in the introduction. Hyphen between p97/VCP-mediated clearance.

Revised.

(d) The following sentence in the results is duplicated: “We considered that direct post-translational modification (PTM) of RNF168 could represent one possible mechanism to regulate its activity and mined publicly available “eukaryotic linear motif (ELM) resource” 30 for potential RNF168 regulatory motifs.”

Apologies, revised.

(e) “numerous potential PIN1 binding sites near”, should be PIN1-binding sites (also in Fig S1A, and elsewhere in the manuscript).

Revised.

(f) The following sentence in the results is duplicated: “In addition, we also observed excessive RNF168 chromatin accumulation both in untreated and IR-treated cells incubated with small molecule inhibitors of PIN1 that either affect its enzymatic activity (Juglone 31 and PiB 32) or its stability (ATRA 33) (Figure 1I & J, Supplementary Figure 1C).”

Apologies, revised.

(g) Please check the legend on the y-axis in Fig 1I/J, Fig 3E/F and Fig 5F: “intensity per focal plan”. I assume focal plane is meant here?

Revised.

(h) Results: “used these to test purification of RNF168 from cell extracts” rephrase to clarify. For instance, used these to co-immunoprecipitate RNF168 from cell extracts.

Sentence removed.

(i) Results: “We observed increased PIN1 and RNF168204-210 mass products ..” annotate RNF168204-210 differently for clarity. Perhaps 204-2010-RNF168, or RNF168 (204-2010)?

Revised.

(j) The following sentence in the results is duplicated: “The modification of RNF168 by SUMO1 is known to be mediated by PIAS4, and has been implicated in promoting its transcription 45.”

Revised.

(k) Results: “we depleted SUMO1 and SUMO2 with SUMO3”. Rephrase to clarify. Perhaps we depleted SUMO1 and SUMO2/3?

Revised.

(l) Discussion: “clearance of RNF168 appearsn to influence”, correct to “appears”.

Revised.

References

1. Cerovsky, V. and H.A. Scheraga, *Combined solid-phase/solution synthesis of large ribonuclease A C-terminal peptides containing a non-natural proline analog*. J Pept Res, 2005. **65**(6): p. 518-28.
2. Nakamura, K., et al., *Proline isomer-specific antibodies reveal the early pathogenic tau conformation in Alzheimer's disease*. Cell, 2012. **149**(1): p. 232-44.
3. Meerang, M., et al., *The ubiquitin-selective segregase VCP/p97 orchestrates the response to DNA double-strand breaks*. Nature cell biology, 2011. **13**(11): p. 1376-82.
4. Kilgas, S., et al., *p97/VCP inhibition causes excessive MRE11-dependent DNA end resection promoting cell killing after ionizing radiation*. Cell Rep, 2021. **35**(8): p. 109153.
5. Thorslund, T., et al., *Histone H1 couples initiation and amplification of ubiquitin signalling after DNA damage*. Nature, 2015. **527**(7578): p. 389-93.
6. Richards, F., et al., *Regulation of Rad52-dependent replication fork recovery through serine ADP-ribosylation of PolD3*. Nat Commun, 2023. **14**(1): p. 4310.
7. Kelliher, J.L., et al., *Evolved histone tail regulates 53BP1 recruitment at damaged chromatin*. Nat Commun, 2024. **15**(1): p. 4634.
8. Kuster, A., et al., *A stapled peptide mimetic of the CtIP tetramerization motif interferes with double-strand break repair and replication fork protection*. Sci Adv, 2021. **7**(8).